# Unveiling the Enigmatic nature of six neglected Amazonian *Leishmania* (*Viannia*) species using the hamster model: Virulence, Histopathology and prospection of LRV1

**Rodrigo Pedro Soares**[1,2]*, **Igor Campos Fontes**[1], **Felipe Dutra-Rêgo**[1], **Jeronimo Nunes Rugani**[1], **Paulo Otávio L. Moreira**[1], **Vânia Lúcia Ribeiro da Matta**[2], **Gabriela Venícia Araujo Flores**[2], **Carmen Maria Sandoval Pacheco**[2], **Andrey José de Andrade**[3], **Magda Clara Vieira da Costa-Ribeiro**[3], **Jeffrey Jon Shaw**[4], **Márcia Dalastra Laurenti**[2]*

**1** Grupo Biotecnologia Aplicada ao Estudo de Patógenos (BAP), Instituto René Rachou, Fundação Oswaldo Cruz (FIOCRUZ), Belo Horizonte, Minas Gerais, Brazil, **2** Laboratório de Patologia das Moléstias Infecciosas, Departamento de Patologia, Faculdade de Medicina, Universidade de São Paulo (USP), São Paulo, São Paulo, Brazil, **3** Laboratório de Parasitologia Molecular, Departamento de Patologia Básica, Setor de Ciências Biológicas, Universidade Federal do Paraná (UFPR), Curitiba, Paraná, Brazil, **4** Departamento de Parasitologia, Instituto de Ciências Biomédicas, Universidade de São Paulo (USP), São Paulo, São Paulo, Brazil

\* rodrigosoares28@hotmail.com (RPS); mdlauren@usp.br (MDL)

**Data Availability Statement:** The authors confirm that all data underlying the findings are fully

## Abstract

American tegumentary leishmaniasis (ATL) is highly endemic in the Amazon basin and occurs in all South American countries, except Chile and Uruguay. Most Brazilian ATL cases are due to *Leishmania (Viannia) braziliensis*, however other neglected Amazonian species are being increasingly reported. They belong to the subgenus *L. (Viannia)* and information on suitable models to understand immunopathology are scarce. Here, we explored the use of the golden hamster *Mesocricetus auratus* and its macrophages as a model for *L. (Viannia)* species. We also studied the interaction of parasite glycoconjugates (LPGs and GIPLs) in murine macrophages. The following strains were used: *L. (V.) braziliensis* (MHOM/BR/2001/BA788), *L. (V.) guyanensis* (MHOM/BR/85/M9945), *L. (V.) shawi* (MHOM/BR/96/M15789), *L. (V.) lindenbergi* (MHOM/BR/98/M15733) and *L. (V.) naiffi* (MDAS/BR/79/M5533). *In vivo* infections were initiated by injecting parasites into the foot-pad and were followed up at 20- and 40-days PI. Parasites were mixed with salivary gland extract (SGE) from wild-captured *Nyssomyia neivai* prior to *in vivo* infections. Animals were euthanized for histopathological evaluation of the footpads, spleen, and liver. The parasite burden was evaluated in the skin and draining lymph nodes. *In vitro* infections used resident peritoneal macrophages and THP-1 monocytes infected with all species using a MOI (1:10). For biochemical studies, glycoconjugates (LPGs and GIPLs) were extracted, purified, and biochemically characterized using fluorophore-assisted carbohydrate electrophoresis (FACE). They were functionally evaluated after incubation with macrophages from C57BL/6 mice and knockouts (TLR2-/- and TLR4-/-) for nitric oxide (NO) and cytokine/chemokine production. All species, except *L. (V.) guyanensis*, failed to generate evident macroscopic lesions 40 days PI. The *L. (V.) guyanensis* lesions were swollen but did not ulcerate and

available without restriction. All relevant data are within the paper and its Supporting Information files.

**Funding:** This work was supported by Conselho Nacional de Desenvolvimento Científico e Tecnológico (305238/2023-0 to RPS, 308817/2021-4 to MDL); the Fundação de Amparo à Pesquisa do Estado de Minas Gerais (PPM-XII 00202-18 to RPS); and the Fundação de Amparo do Estado de São Paulo (2020/05388-0 to MDL and 2021/01243-0 to RPS). No authors received salary from the founders. The funders had no role in study design, data collection and analysis, decision to publish, or preparation of the manuscript.

**Competing interests:** The authors have declared that no competing interests exist.

microscopically were characterized by an intense inflammatory exudate. Despite the fact the other species did not produce visible skin lesions there was no or mild pro-inflammatory infiltration at the inoculation site and parasites survived in the hamster skin/lymph nodes and even visceralized. Although none of the species caused severe disease in the hamster, they differentially infected peritoneal macrophages *in vitro*. LPGs and GIPLs were able to differentially trigger NO and cytokine production via TLR2/TLR4 and TLR4, respectively. The presence of a sidechain in *L. (V.) lainsoni* LPG (type II) may be responsible for its higher proinflammatory activity. After Principal Component analyses using all phenotypic features, the clustering of *L. (V.) lainsoni* was separated from all the other *L. (Viannia)* species. We conclude that *M. auratus* was a suitable *in vivo* model for at least four dermotropic *L. (Viannia)* species. However, *in vitro* studies using peritoneal cells are a suitable alternative for understanding interactions of the six *L. (Viannia)* species used here. LRV1 presence was found in *L. (V.) guyanensis* and *L. (V.) shawi* with no apparent correlation with virulence *in vitro* and *in vivo*. Finally, parasite glycoconjugates were able to functionally trigger various innate immune responses in murine macrophages *via* TLRs consistent with their inflammatory profile *in vivo*.

## Author summary

American Tegumentary Leishmaniasis (ATL) is one of the major neglected diseases occurring in Brazil and the Amazon contributes with a wide range of species that are not completely known. These little-known species are transmitted to rural workers entering the forest during lumbering, mining, and latex extraction activities and rarely in peri domestic situation. Infection occurs by the bite of infected sand flies in different environments and biomes. One of the major constraints in studying *L. (Viannia)* species is the difficulty of finding suitable animal and cellular models to investigate their biology. The mouse model is not a suitable model for most *L. (Viannia)* species. However, hamsters were considered a good model for *L. (V.) braziliensis*. In the search for a better model for some neglected dermotropic Amazonian *Leishmania*, we evaluated interactions *in vivo* in the hamster and *in vitro* in its macrophages. All species infected cells *in vitro*, but hamsters could only be considered as a suitable *in vivo* model for *L. (V.) guyanensis*. Importantly the glycoconjugates of the six *L. (Viannia)* species were very proinflammatory, showing various levels of macrophage activation. This paper opens possibilities of determining more accurately the taxonomic position of *L. (V.) lainsoni* within the subgenus *L. (Viannia)*.

## Introduction

*Leishmania* species from *L. (Viannia)* subgenus have been identified from patients exhibiting American Tegumentary leishmaniasis (ATL) [1]. Most Brazilian ATL cases are caused by *Leishmania (Viannia) braziliensis*, and in past years [2] there has been an increase in the number of cases. Eight species are considered proven vectors including *Nyssomyia intermedia* and *Ny. whitmani* [3]. Several wild and synanthropic reservoirs have been identified including rodents [4–6] and marsupials [7,8]. Recently, urbanization of ATL was reported in the Southeast Brazil, where *L. (V.) braziliensis* proven vectors *Migonemyia migonei*, *Ny. intermedia* and

*Ny. whitmani* were captured [9,10]. This novel finding sparked concern in the disease epidemiology since urbanization is associated to visceral leishmaniasis (VL) due to *L. (Leishmania) infantum* [11].

*L. (V.) braziliensis* and *L. (V.) guyanensis* are epidemiologically the most important ATL species. Most cases are in the north and northeast of Brazil, where *L. (Viannia)* parasites are endemic, the disease being a zoonosis and occupational hazard [12]. *Nyssomyia umbratilis*, found in the Amazon region and other Brazilian states, transmits *L. (V.) guyanensis* above the Negro River [13,14]. The frequent infection of this species by *Leishmania* RNA virus 1 (LRV1), a *Totiviridae* virus known to increase disease severity, is a distinguished characteristic of *L. guyanensis* [15,16]. Other dermotropic species also contribute to ATL increase in Latin America. These include *L. (V.) naiffi*, *L. (V.) shawi*, *L. (V.) lindenbergi* and *L. (V.) lainsoni*. *L. (V.) naiffi* was first isolated from an armadillo (*Dasypus novemcinctus*) in the Brazilian state of Pará, Colombia and French Guyana [17,18]. The suspected vectors were *Psychodopygus paraensis* and *Ps. ayrozai* [19]. Recently, *L. (V.) naiffi* DNA was detected in several sandfly species including *Ps. hirsutus*, *Ny. antunesi* and *Ny. shawi* in the Brazilian state of Rondônia [20]. Recently, *L. (V.) lainsoni* has also been detected in French Guyana, Peru and Bolivia [18], but its first description was from a human case in Pará state, Brazil [21], where *Cuniculus paca* (Rodentia: Cuniculidae) was incriminated as the wild reservoir [22]. Later, field-captured *Trichophoromyia ubiquitalis* and *Trichophoromyia brachipyga* were incriminated as suspected and proven vectors of *L. (V.) lainsoni*, respectively [23]. A distinguished feature of *L. (V.) lainsoni* compared to the other *L. (Viannia)* species is that it possesses several biological, biochemical, and molecular characteristics that differ from the others of the same subgenus [24]. *L. (V.) shawi* was also reported in Pará state, Brazil, whose wild reservoirs include *Cebus apella*, *Chiropotes satanas*, *Choloepus didactylus*, *Bradypus tridactylus*, and *Nasua nasua* [25]. Later, this species was isolated from human cases and the suspected vector was *Ny. whitmani* [26]. Finally, *L. (V.) lindenbergi* was first reported in an ATL outbreak among eight soldiers in Pará state, Brazil. Clinically it is very similar to *L. (V.) naiffi*, whose suspected vector is *Ny. antunesi* and sylvatic reservoirs are unknown [27]. Recently, *L. (V.) lindenbergi* was also reported in another Brazilian state of Rondonia where *Ny. antunesi* occurs [20,28]. Several factors are responsible for ATL increase in the Americas including climate, economic, environmental and others [29]. Those are neglected species and even though some aspects of their biology are known, information on suitable animal models is still scarce.

For over 60 years [30] the hamster has been a model for several pathogens. This model is suitable for both viscerotropic and dermotropic *Leishmania* and its clinicopathologic features and immunopathologic mechanisms are like those found in humans, but quite different from mice [31–33]. Hamsters have been considered an acceptable model for *L. (V.) braziliensis* infection [34] but we do not know their susceptibility to infection with other less common dermotropic *L. (Viannia)* species. In this work, in the search for a better model and to understand immunopathology, we evaluated the interaction of *L. (Viannia)* species and their glycoconjugates (LPGs and GIPLs) with *M. auratus* and human/rodent macrophages.

## 2. Material and methods

### Ethics statement

Animals (*Mesocricetus auratus*) were kept in the Animal Facility of the Faculdade de Medicina da Universidade de São Paulo (FMUSP). *Mus musculus* (C57BL/6, TLR2 -/- and TLR4 -/-) were kept in the Animal Facility of Instituto René Rachou, Oswaldo Cruz Foundation. Animal handling was performed according to the guidelines of the Brazilian College for Experiments with Animals (COBEA-Law 11.794/2008). Our protocols were approved by our Ethics

**Table 1.** *Leishmania* strains used in this study.

| Strain | WHO reference code | Nomenclature in the text | Host | Origin |
|---|---|---|---|---|
| *Leishmania (V.) guyanensis* | MHOM/BR/75/M4147 | M4147 | *Homo sapiens* | Pará/BR |
| *Leishmania (V.) guyanensis* | IUMB/BR/85/M9945 | M9945 | *Ny. umbratilis* | Pará/BR |
| *Leishmania (V.) braziliensis* | MHOM/BR/01/BA788 | BA788 | *Homo sapiens* | Bahia/BR |
| *Leishmania (V.) shawi* | MHOM/BR/96/M15789 | M15789 | *Homo sapiens* | Bahia/BR |
| *Leishmania (V.) lainsoni* | MHOM/BR/81/M6426 | M6426 | *Homo sapiens* | Bahia/BR |
| *Leishmania (V.) lindenbergi* | MHOM/BR/1996/M15733 | M15733 | *Homo sapiens* | Bahia/BR |
| *Leishmania (V.) naiffi* | MDAS/BR/1979/M5533 | M5533 | *Dasypus* sp | Minas Gerais/BR |
| *Leishmania (L.) infantum* | MCAN/BR/89/Ba-262 | BA262 | *Canis familiaria* | Bahia/BR |

Committee in Animal Experimentation/Comitê de Ética em Experimentação Animal (ECAE/ CEUA) (Protocol 1659/2021 and LW-32/16).

## Parasite culture and molecular typing

Three isolates and three World Health Organization Reference *L.* (*Viannia*) species were evaluated (Table 1). Promastigotes were cultured in M199 medium and/or Schneider supplemented with 10% fetal bovine serum (FBS) (Invitrogen/ Thermo Fisher Scientific (Carlsbad, USA), penicillin (200 u/mL), and streptomycin (200 µg/mL) (all Merck KGaA, Darmstadt, Germany), at 25°C [35]. To confirm parasite identity, molecular typing (*hsp*70 gene) was performed [36]. Confirmed sequences were deposited in the GenBank database (accession numbers PP331240, PP331241, PP331242, PP331243, PP331244 and PP331245).

## Extraction and purification of glycoconjugates

Lipophosphoglycans (LPGs) and glycoinositolphospholipids (GIPLs) were extracted and purified using Solvent E in a hydrophobic interaction column as previously reported [37]. Purified LPGs and GIPLs were depolymerized using mild and strong acid hydrolysis, respectively. Poly- (repeat units) and monosaccharides were resolved by fluorophore-assisted carbohydrate electrophoresis (FACE) and the profiles were visualized under UV [38].

## *In vitro* assays

Hamster resident intraperitoneal macrophages were recovered by sequential washing with 10 mL of cold PBS. Cell viability was checked by trypan blue. THP-1 cells were differentiated by addition of 20 nM phorbol 12-myristate 13-acetate as previously reported [39]. Cells ($10^5$/well) were seeded into 24-well culture plates for adhesion (1 h, 35°C, 5% $CO_2$) with RPMI supplemented with 2 mM glutamine, 50 U/mL of penicillin, 50 µg/mL streptomycin and 10% FBS. Cells were exposed to *L. Viannia* species ($10^6$/well, 4 h, MOI 1:10). After 48 hours, 13-mm round coverslips were recovered and stained with Panoptic. For functional studies with glycoconjugates, thioglycolate-elicited macrophages were removed from mice peritoneal cavity (C57BL/6) and respective knockouts (TLR2 -/- and TLR4 -/-). Macrophages were attached to 96-well plates and primed with IFN-γ (3U/mL) for 16 h prior to incubation with LPG and GIPLs (10 µg/mL) for 48 h (37°C, 5% $CO_2$). Negative control was RPMI medium only. Positive controls were primed with IFN-γ (3U/mL) and included LPS (100 ng/mL, TLR4 agonist) and *Staphylococcus aureus* extract (100 ng/mL, TLR2 agonist) [40]. Supernatants were assayed for nitric oxide and cytokines using Griess method and CBA flex Inflammation kit (Beckton and Dickinson) according to manufacturer's instructions. The following cytokines/chemokine were assayed TNF-α, IL-6, IL-12, and MCP-1. Two parallel experiments were performed in

triplicate totalizing 6 samples for each glycoconjugate exposure. These experiments were performed in the Flow Cytometry platform in the Rene Rachou Institute, Oswaldo Cruz Foundation.

## Sand flies and salivary gland dissection

*Nyssomyia neivai* is a suspected vector of *L.* (*V.*) *braziliensis* in the South of Brazil [41]. The ability of this sand fly to acquire infection from *L.* (*V.*) *braziliensis*-infected hamsters has already been reported [42]. Salivary glands extracts (SGEs) are important modulators of early and late infection [43]. For this reason, we chose this species´ salivary glands for our studies with *L.* (*Viannia*) isolates. Sand flies found resting on the wall of a residence located in Castro municipality (24˚59'31.2" S, 49˚37'40.8" W), Paraná, Brazil were captured with an aspirator. They were placed in a bottle with pieces of cotton soaked in a 10% sucrose solution prior to dissection in the laboratory. *Ny. neivai* females' SGEs were dissected in PBS under a stereomicroscope and stored. *Vouchers* were deposited in the "Coleção de Parasitologia do Departamento de Patologia Básica" (ColPar/DPAT/UFPR).

## *In vivo* assays

For each species, 8 hamsters were inoculated subcutaneously (SC) in the right rear footpad with 1 X $10^5$ of stationary parasites in a volume of 100 μL of PBS and salivary gland extract (0.5 gland) of *Ny. neivai*. Negative controls were inoculated with PBS only. Animals were followed by 20- and 40-days PI. At the two time-points, hamsters were anaesthetized and euthanized. Skin, liver, and spleen fragments were collected for histopathology (hemotoxylin-eosin-HE) and immunohistochemistry (IHC) as previously reported [43]. To evaluate the inflammation in the skin, a semi-quantitative comparative analysis was performed [44]. The results were scored based on the intensity of cellularity as follows: (−) negative, 1–50 cells (+) discrete, 50–100 cells (++) moderate and higher than 100 cells (+++) intense. To determine parasite load, limiting dilution was performed in the skin and draining lymph nodes as reported elsewhere [45].

## LRV1 detection

Attempts to detect LRV1 were performed using RNA extracted from cultured parasites (1 x $10^8$ cells) using the Trizol method (Life Technologies). Samples were treated with DNase I before cDNA synthesis as previously reported, using the Super Script III- First Strand Synthesis Kit (Invitrogen). Specific primers (SMB2109 and SMB2110) were used to amplify a 496 bp LRV1 fragment and β-tubulin (396 bp) was used as quality control. Amplified fragments were resolved in 1% agarose gels [46,47].

## Statistical and Principal Component analyses (PCA)

For morphometric semi-quantitative analysis, 25 fields of each slide were randomly photographed. Quantification of parasitism and mononuclear cells for each species were performed in KS300 software. Data were analyzed using GraphPad Prism 5.0. Statistical analyses used One-way ANOVA followed by Bonferroni´s and Dunn´s Multiple Comparison tests. For NO and cytokines/chemokine, the Shapiro Wilk test was conducted to test the null hypothesis that data were sampled from a Gaussian distribution [48]. When data did not deviate from a Normal distribution, Student´s "t" test were performed. *P*-values <0.05 were considered statistically significant. Principal Component Analysis (PCA) and hierarchical clustering by squared-Euclidean distance [33] were performed using the software IBM SPSS Statistics 22. This approach used all biological, biochemical, and immunological parameters summarized in

**Table 2. Biological parameters assessed in *Leishmania* strains used in this study.**

| Strain | *In vitro* infection for macrophages | | *In vivo* infection in *M. auratus* | | | | LRV1 |
|---|---|---|---|---|---|---|---|
| | Hamster | THP-1 | HE | IHC | LD 20/40 days PI | | |
| | infection/amastigotes per cell | infection/amastigotes per cell | 20/40 days PI | 20/40 days PI | Skin | Lymph node | |
| *L. (V.) guyanensis* (M9945) | +++/+++ | ++/++ | +++/- | +/+++ | +++/+++ | ++/+++ | +++ |
| *L. (V.) shawi* (M15789) | +++/++ | +/+ | -/- | +/++ | -/+++ | +/+++ | ++ |
| *L. (V.) lainsoni* (M6426) | ++/++ | ++/++ | +/- | ++/++ | +++/++ | +++/+++ | - |
| *L. (V.) braziliensis* (BA788) | ++/++ | +/+ | +++/++ | +/+++ | ++/+++ | +++/+++ | - |
| *L. (V.) naiffi* (M5533) | +/+ | +/+ | -/- | +/- | -/- | -/- | - |
| *L. (V.) lindenbergi* (M15733) | +/+ | ++/+ | -/- | +/+ | -/- | -/- | - |

+++, higher infection; ++, moderate infection; +, lower infection; -, not detectable; HE, hematoxylin-eosin staining; IHC, immunohistochemistry; LD, limiting dilution; PI, post-infections and, LRV1, *LeishmaniaRNAvirus* 1.

Tables 2 and 3 to generate an integrative description of the different characteristics of the *L. (Viannia)* species.

## Results

### LRV1 presence did not seem to affect infection in hamster resident peritoneal macrophages and human THP-1 cells

All strains had variable levels of infectivity with resident macrophages of *M. auratus* (Fig 1). *L. (V.) guyanensis* and *L. (V.) shawi* had the highest infection levels of macrophages (>50%). *L. (V.) braziliensis* and *L. (V.) lainsoni* had moderate infection levels (~40%) whereas *L. (V.) lindenbergi/L. (V.) naiffi* were the lowest (<6%) (Fig 2A, P<0.05). In respect to amastigote intracellular parasitism *L. (V.) guyanensis* had the highest number of parasites (~10/cell) (P<0.05). The remaining species exhibited lower numbers of amastigotes ranging from 1.6–1.8 in *L. (V.) lindenbergi/L. (V.) naiffi* (P>0.05) to 3.4–5.3 for *L. (V.) shawi/L. (V.) braziliensis/L. (V.) lainsoni* (Fig 2B) (P>0.05). All strains were searched for *Leishmania* RNA virus 1 (LRV1), because

**Table 3. Biochemical and Functional properties of *Leishmania* glycoconjugates used in this study.**

| Parameter | Strains/Glycoconjugates | | | | | | | | | | | |
|---|---|---|---|---|---|---|---|---|---|---|---|---|
| | L.guy (M9945) | | L.sha (M15789) | | L.lai (M6426) | | L.braz (BA788) | | L.nai (M5533) | | L.lin (M15733) | |
| | LPG | GIPL | LPG | GIPL | LPG | GIPL | LPG | GIPL | LPG | GIPL | LPG | GIPL |
| Type | I | II/Hybrid | I | I | I | I | II | I | I | I | I | I |
| NO/TLR2 | + | + | ++ | ++ | ++ | + | + | + | + | + | ++ | + |
| NO/TLR4 | ++ | ++ | ++ | ++ | +++ | +++ | ++ | ++ | ++ | ++ | ++ | ++ |
| IL-6/TLR2 | - | + | + | ++ | - | +++ | - | + | - | + | +++ | + |
| IL-6/TLR4 | + | - | + | + | +++ | +++ | + | + | + | + | + | + |
| IL-12/TLR2 | - | + | - | - | - | ++ | - | - | - | - | - | - |
| IL-12/TLR4 | ++ | + | + | ++ | + | - | + | ++ | + | - | + | - |
| TNF-α/TLR2 | - | - | ++ | ++ | +++ | +++ | - | - | - | - | - | - |
| TNF-α/TLR4 | + | - | ++ | ++ | + | +++ | + | + | + | + | + | + |
| MCP-1/TLR2 | + | - | + | - | + | - | + | - | + | - | + | - |
| MCP-1/TLR4 | + | + | + | + | +++ | + | ++ | + | ++ | + | ++ | + |

**Legend:** LPG, Lipophosphoglycan; GIPLs, glycoinositolphospholipids; TLR2, toll-like receptor 2; TLR4, toll-like receptor 4; NO, nitric oxide; IL-6, interleukin 6; IL-12, interleukin 12; TNF-α, tumor necroses factor alpha and, MCP-1, monocyte chemoattractant protein-1.

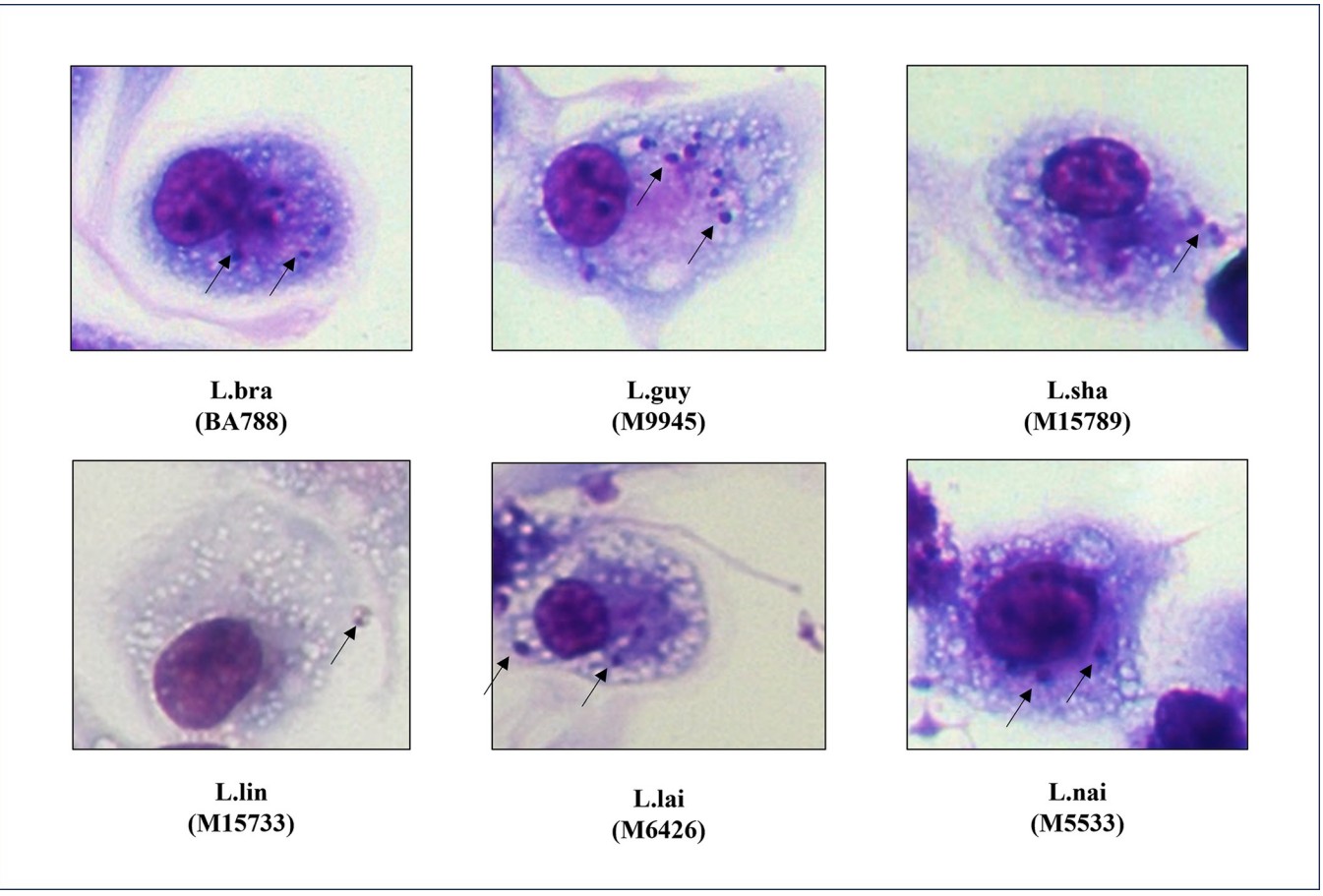

**Fig 1. Resident peritoneal macrophages from *Mesocricetus auratus* are susceptible to *in vitro Leishmania* (*Viannia*) infection.** Arrows indicate intracellular amastigote forms. Legend: L.braz, *L. braziliensis*; L.guy, *L. guyanensis*; L.sha, *L. shawi*, L.lind, *L. linderbergi*, L.lai, *L. lainsoni* and L.nai, *L. naiffi*. Magnification of 1000x.

its presence was related with *Leishmania* virulence. The reference strain of *L.* (*V.*) *guyanensis* (M4147) was positive for LRV1. LRV1 was detected in *L.* (*V.*) *guyanensis* (M9945) and in *L.* (*V.*) *shawi* (M15789) (Fig 2C) (Table 2). β-tubulin gene confirmed DNA integrity of all strains (Fig 2D). THP-1 cells had a lower infection percentage (11–25%) and number of intracellular amastigotes per cell (1–2) in comparison to hamster resident macrophages (S1A–S1C Fig).

### *Leishmania* (*Viannia*) *guyanensis* caused macroscopical lesions in hamsters

Since all strains infected hamster macrophages *in vitro*, the next step was to perform *in vivo* infections of the parasites in the presence of SGEs of *Ny. neivai*. Except for *L.* (*V.*) *guyanensis*, all strains failed to cause macroscopic lesions in the foot pad of the hamsters (Fig 3). Although no ulceration was seen, an increase in the swelling of the foot pad was observed after 20 days PI reaching its peak at 40 days PI (Fig 3A) (P<0.05). A representative image of two animals shows right feet pads swollen in comparison to left ones (Fig 3B).

### *L.* (*Viannia*) species exhibited different levels of pathological severity

After 20- and 40-days PI, hamsters were anesthetized and euthanized prior to foot pad, liver, and spleen removal. Although no external macroscopical ulceration was detected in skin, an

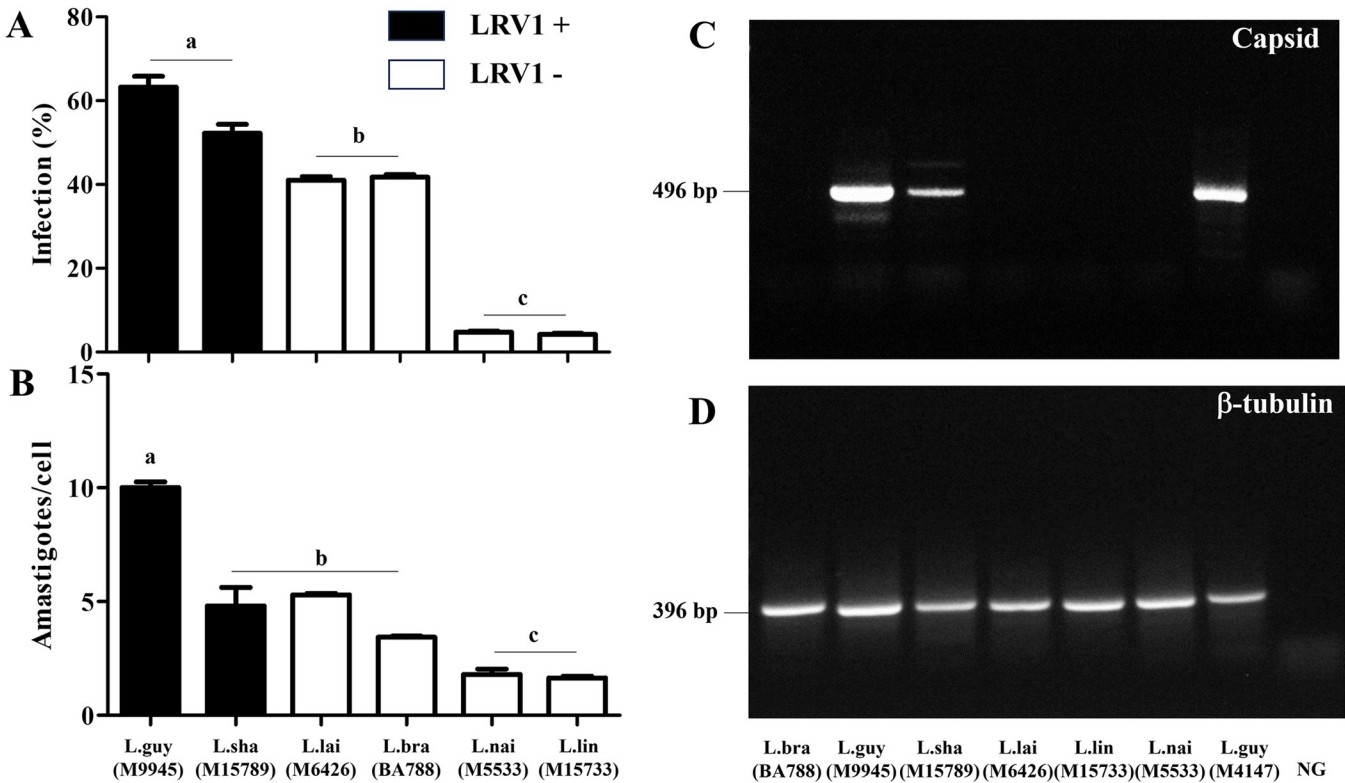

**Fig 2. *Leishmania* species shows different levels of infection in resident peritoneal macrophages of *M. auratus* in the presence/absence of *Leishmania* RNA virus 1 (LRV1).** (A) Macrophage infection (%), (B) number of amastigotes per macrophage, (C) LRV1 detection based on amplification of capsid gene and (D) DNA integrity control based on amplification of β-tubulin gene. Legend: L.braz, *L. braziliensis*; L.guy, *L. guyanensis*; L.sha, *L. shawi*, L.lind, *L. linderbergi*, L.lai, *L. lainsoni*, L.nai, *L. naiffi* and NG, negative control. Positive control is represented by *L. guyanensis* (L.guy M4147). Dark and white bars indicate LRV1 presence and absence, respectively. The same letters above bars do not indicate statistical differences (P>0.05). Different letters indicate statistical differences (P<0.05).

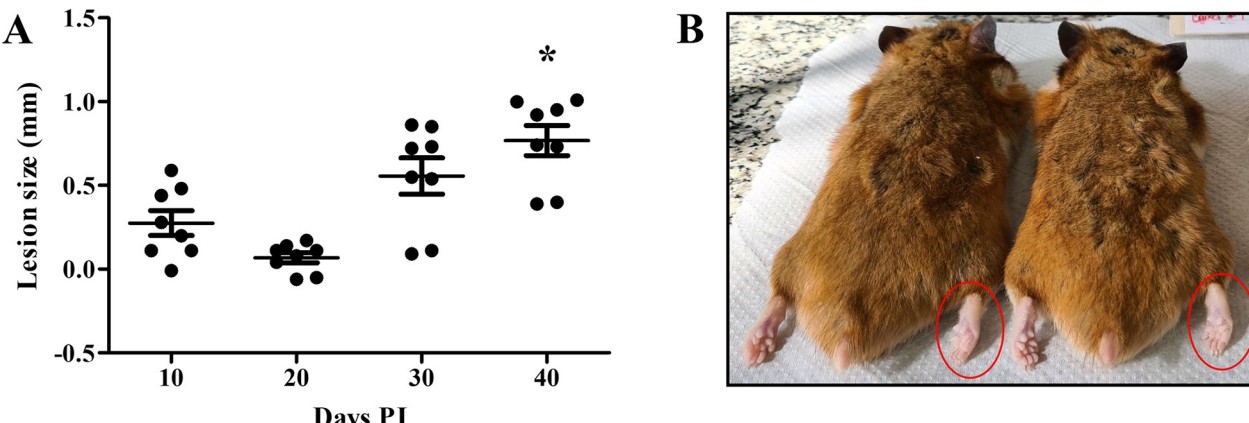

**Fig 3. *Leishmania guyanensis* (M9945) in the presence of salivary gland extract (SGE) caused macroscopic lesion in *M. auratus* foot pads.** (A) Size of the lesion (mm) and (B) macroscopic evaluation of swollen right foot pads (red circles). Asterisk indicates statistical difference in the lesion size between 40- and 10/20- days PI.

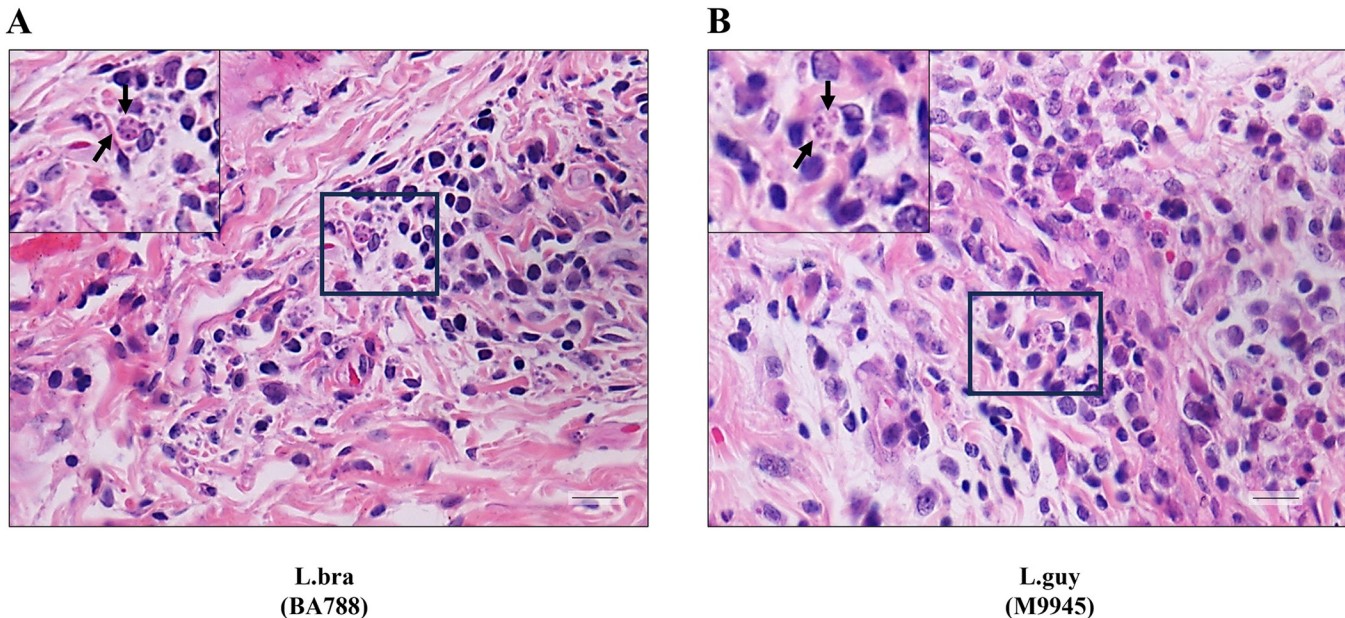

**L.bra
(BA788)**

**L.guy
(M9945)**

**Fig 4. *L. braziliensis* (A) and *L. guyanensis* (B) had higher levels of mononuclear infection in the dermis of *M. auratus* 20 days PI.** Left upper sections in higher magnification showing macrophages infected with amastigote forms (dark arrows). Legend: L.braz, *L. braziliensis* and L.guy, *L. guyanensis*. Scale bar = 20 um.

inflammatory infiltrate was triggered by some species at variable levels. After 20 days PI, this feature was more perceptible for *L.* (*V.*) *braziliensis*, *L.* (*V.*) *guyanensis* and *L.* (*V.*) *lainsoni* in HE stained section. They produced a strong inflammatory infiltration in the presence of infected mononuclear cells containing amastigotes (black arrows, Figs 4A and 4B, and 5). Although *L.* (*V.*) *shawi* also induced areas of proinflammatory infiltration indicated by the asterisks, no amastigotes were detected after 20 days PI with HE (Fig 5). Finally, *L.* (*V.*) *lindenbergi* and *L.* (*V.*) *naiffi* showed low/absent inflammation, with no detected parasites by HE (Fig 5). After 40 days PI, the proinflammatory infiltrate decreased in *L.* (*V.*) *braziliensis*, although amastigotes were still seen inside some cells. For *L.* (*V.*) *guyanensis*, *L.* (*V.*) *shawi* and *L.* (*V.*) *lainsoni*, the inflammatory infiltrate remained but with no detectable amastigotes by HE (Fig 6). Like previous results *L.* (*V.*) *lindenbergi* and *L.* (*V.*) *naiffi* did not show any signs of inflammation in the dermis 40 days PI. As expected, negative controls did not show histopathological changes in the skin, liver, and spleen (S2A-S2C Fig). The quantification of those dermis infiltrates/parasites was performed by morphometry and inflammatory scores were provided (Fig 7). Here, *L.* (*V.*) *braziliensis* had higher proinflammatory scores compared to the other species after 20 days PI. However, statistical difference was only achieved in comparison to *L.* (*V.*) *naiffi* (Fig 7A, P<0.05). However, this pattern changed after 40 days PI, where *L.* (*V.*) *shawi* and *L.* (*V.*) *guyanensis* reached higher scores compared to *L.* (*V.*) *lindenbergi* and *L.* (*V.*) *naiffi* (Fig 7B, P<0.05).

### *Leishmania* (*Viannia*) *braziliensis* visceralized and induced granuloma development in the liver

Together with skin, liver and spleen were also removed for histopathological analysis. We did not find any major alterations in the viscera of all strains, except for *L.* (*V.*) *braziliensis*. This species caused two types of granulomas in the liver (Fig 8). One was composed of epithelioid macrophages and giant cells surrounded by a dense capsule of fibrotic tissue after 20 days PI

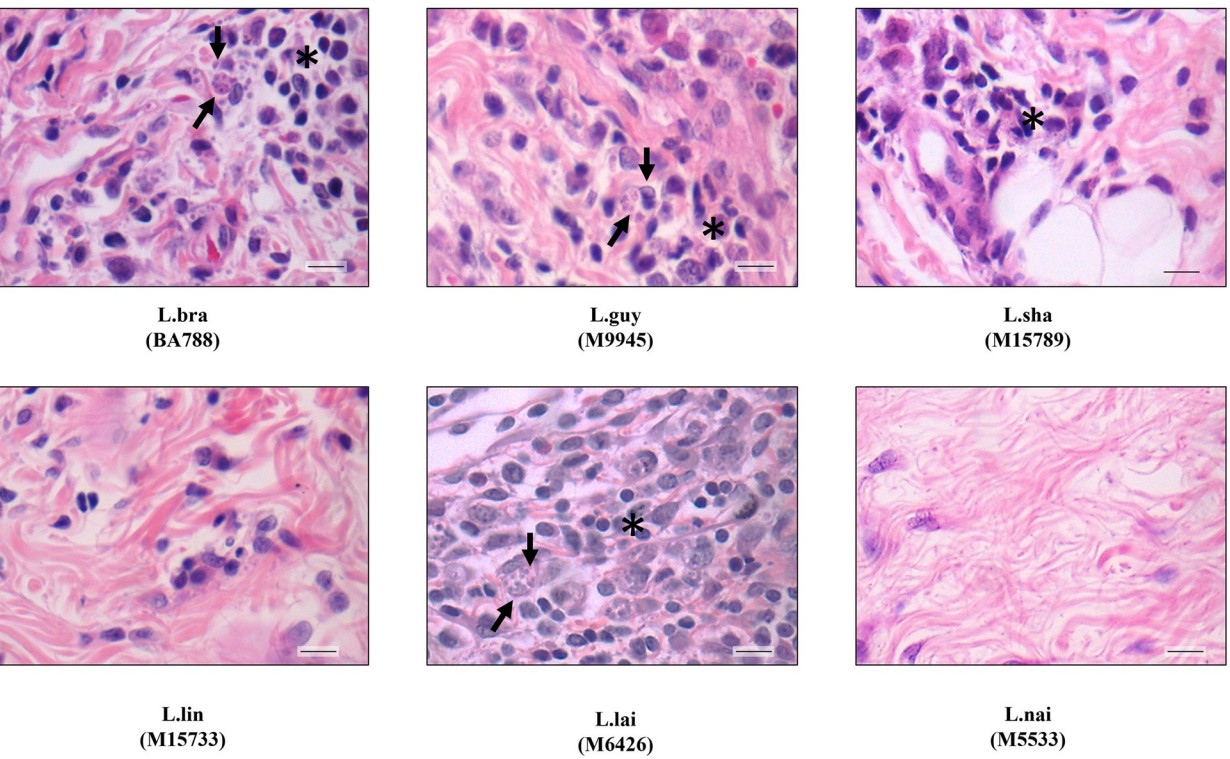

**Fig 5. *Leishmania (Viannia)* species show different levels of pathology in the dermis of *M. auratus* after 20 days PI.** Legend: L.braz, *L. braziliensis*; L.guy, *L. guyanensis*; L.sha, *L. shawi*, L.lind, *L. linderbergi*, L.lai, *L. lainsoni*, L.nai, *L. naiffi*. Scale bar = 20 µm. Dark arrows indicate suggestive intracellular amastigote forms. Asterisks indicate a region with inflammatory infiltrate.

(Fig 8A). The other had a nodular arrangement of inflammatory cells containing lymphocytes and fibroblasts (white arrow) after 40 days PI (Fig 8B). However, no parasites were seen in the surrounding areas of those structures in both periods of PI.

## *L. (Viannia)* species had different levels of parasitism in the dermis of *M. auratus*

Since visible amastigotes were only detected by HE in *L. (V.) braziliensis*, *L. (V.) guyanensis* and *L. (V.) lainsoni* infections, a more sensitive technique, IHC, was used. Unlike HE (Figs 4–6), IHC detected amastigotes in the dermis of animals infected with all species after 20- days PI (Fig 9). After 40 days PI (Fig 10), an increase in the density of parasites was noticed in the dermis of hamsters infected with *L. (V.) braziliensis*/*L. guyanensis* followed by *L. (V.) shawi*/*L. (V.) lainsoni*. A few amastigotes were seen in *L. (V.) lindenbergi* infected hamster, but none were detected in *L. (V.) naiffi* 40 days PI.

We used the limiting dilution assay to improve parasite quantification loads in the foot pad skin and draining lymph nodes. Confirming previous results with HE (Figs 4–6) and IHC (Figs 9–10), we were able to recover parasites from the infected tissues with higher parasite load at both time points (Fig 11A and 11B). In the skin, after 20 days PI, we recovered *L. (V.) braziliensis*, *L. (V.) guyanensis* and *L. (V.) lainsoni*. After 40 days PI, in addition to these same species, we also recovered *L. (V.) shawi*. (Fig 11A). The same recovery pattern exhibited by the skin was repeated in the lymph nodes, except for *L. shawi*, that was also recovered 20 days PI (Fig 11B). In general, parasite load increased with time in both skin and lymph nodes for *L. (V.) braziliensis*/*L. (V.) shawi* and *L. (V.) guyanensis*/*L. (V.) shawi*/*L. (V.) braziliensis*,

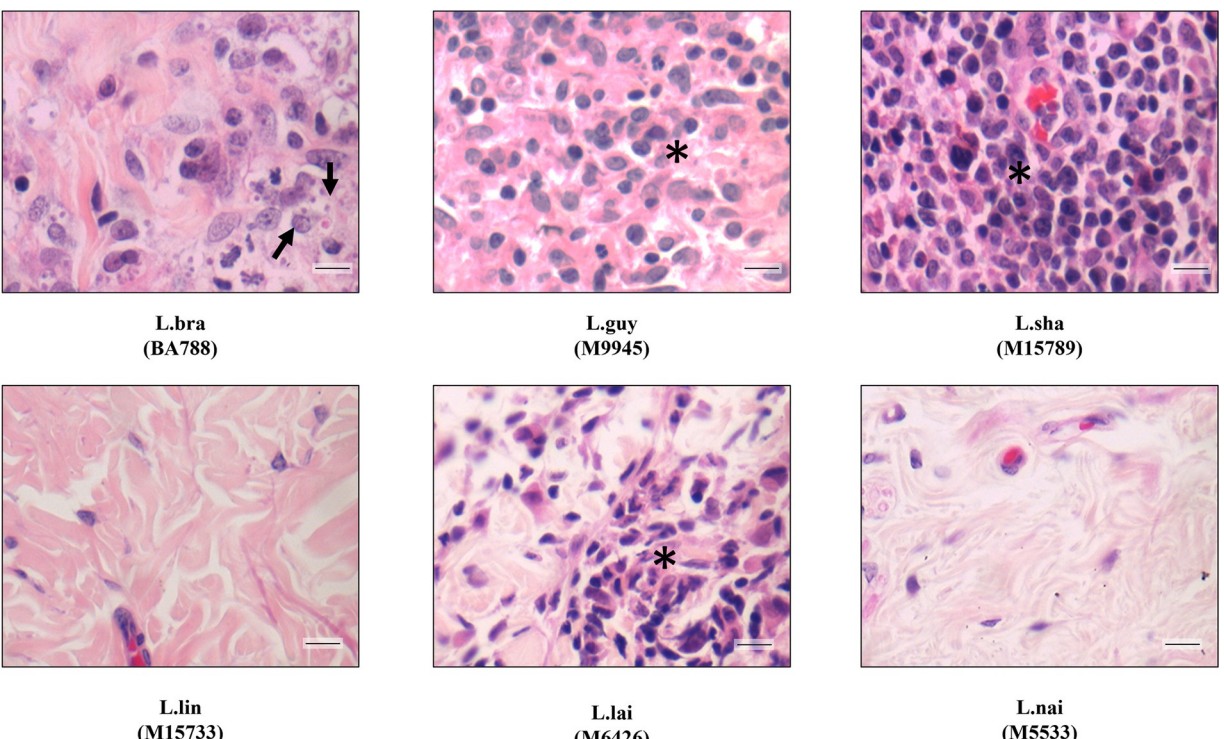

**Fig 6. *In vivo* infection by *Leishmania* (*Viannia*) in the dermis of *M. auratus* shows decrease in parasitism after 40 days PI.** Legend: L.braz, *L. braziliensis*; L.guy, *L. guyanensis*; L.sha, *L. shawi*, L.lind, *L. linderbergi*, L.lai, *L. lainsoni*, L.nai, *L. naiffi*. Scale bar = 20 um. Dark arrows indicate suggestive intracellular amastigote forms. Asterisks indicate a region with inflammatory infiltrate.

respectively (P<0.05). No *L.* (*V.*) *lindenbergi*/*L.* (*V.*) *naiffi* parasites were recovered from any tissue 20- and 40-days PI. All data are summarized in Table 2.

## Glycoconjugates (LPGs and GIPLs) were able to differentially activate murine macrophages *via* TLR4/TLR2

LPG and GIPLs differentially stimulated murine macrophages. Functionally, they were able to induce NO and cytokines/chemokine showing variations in their agonistic proinflammatory

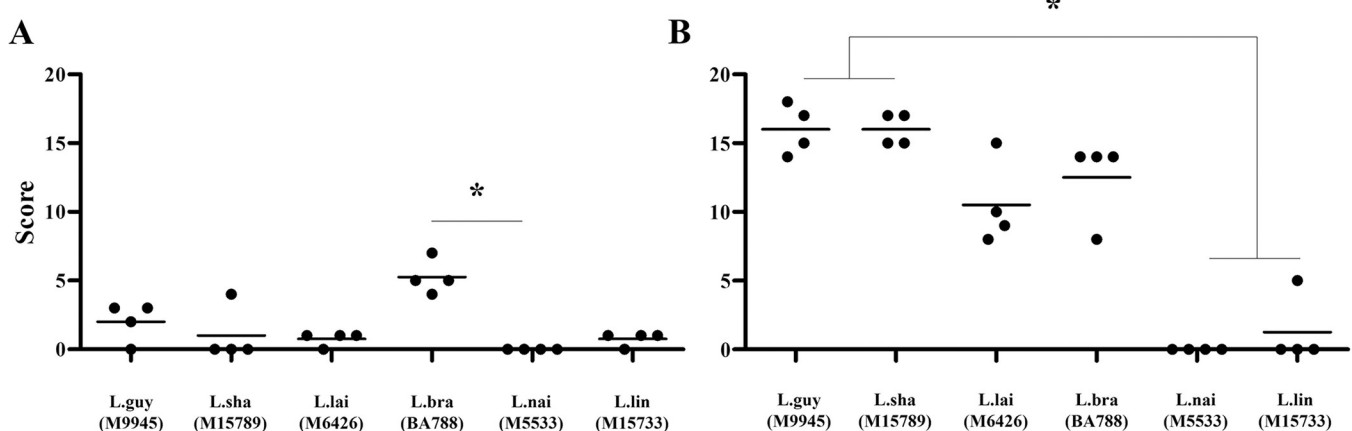

**Fig 7. *Leishmania* (*Viannia*) species display different inflammatory scores in the dermis of *M. auratus*.** (A) 20 days PI and (B) 40 days PI. Legend: L.braz, *L. braziliensis*; L.guy, *L. guyanensis*; L.sha, *L. shawi*, L.lind, *L. linderbergi*, L.lai, *L. lainsoni*, L.nai, *L. naiffi*. Asterisk indicates statistical differences (P<0.05).

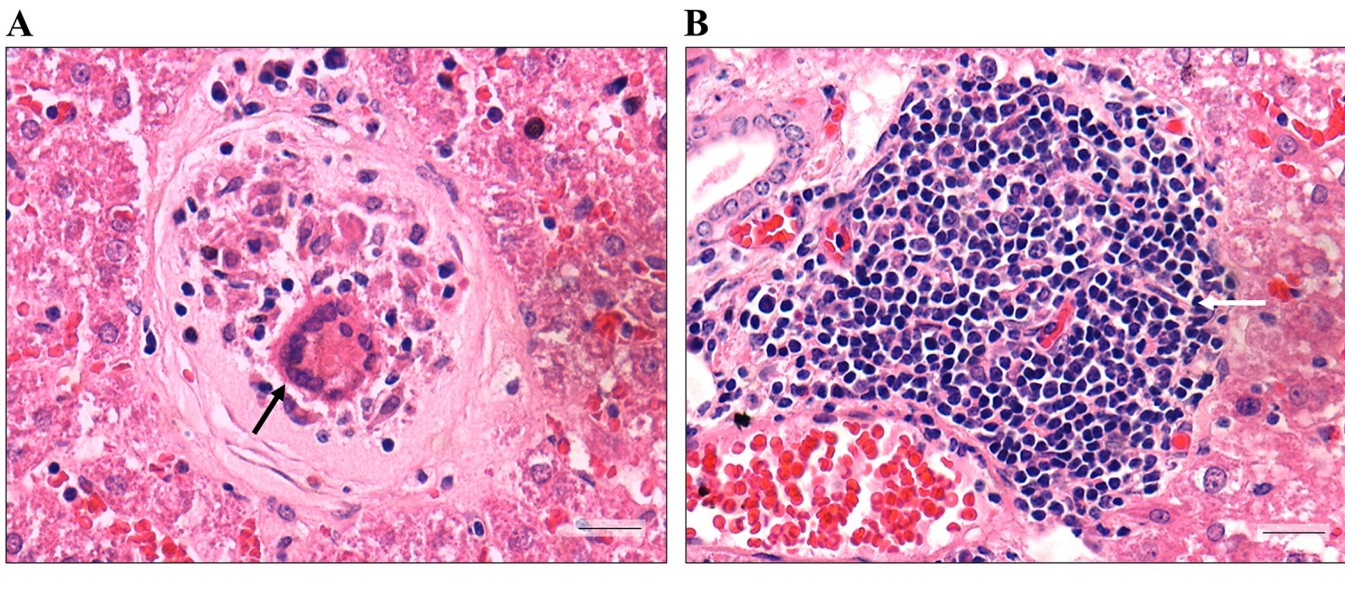

**L.bra**
**(BA788)**

**Fig 8. Visceralization of *L. braziliensis* in *M. auratus* resulted in granuloma-like structures in the liver.** (A) Fibrotic granuloma composed of epithelioid macrophages and giant cells (dark arrow) after 20 days PI. (B) Granuloma composed of lymphoid cells and fibroblasts (white arrow) after 40 days PI. Scale bar = 20 um.

activities. LPGs from all *L.* (*Viannia*) species induced NO *via* TLR4/TLR2, and this ability varied within the three mice lineages. In WT and TLR4 (-/-), LPGs from *L.* (*V.*) *shawi* and *L.* (*V.*) *lindenbergi* induced higher levels of NO, whereas for TLR2 (-/-), *L.* (*V.*) *lainsoni* had higher pro-inflammatory activity (Fig 12A, P<0.05). For GIPLs, a similar pattern was observed (Fig 12B). GIPLs from *L.* (*V.*) *shawi* induced a significant level of NO compared to all species in WT and TLR4 (-/-) (P<0.05). However, for TLR2 (-/-), higher levels of NO were detected in all species, but the highest production was *L.* (*V.*) *lainsoni* (Fig 12, P<0.05). As expected, for WT, both positive controls (LPS and *S. aureus* extract) induced higher levels of NO and cytokines/chemokine (P<0.05). For TLR2 (-/-), LPS induced higher levels of NO, whereas *S. aureus* extract did not. On the other hand, for TLR4 (-/-), LPS induced lower levels of NO compared to *S. aureus* extract (Figs 12–16, P<0.05). For the cytokines/chemokine, in general, LPGs had an agonistic activity *via* TRL4 and secondarily *via* TLR2. On the other hand, GIPLs had a more pro-inflammatory activity *via* TLR4. Sometimes, glycoconjugates frow *L.* (*V.*) *shawi* also had considerable proinflammatory activity, especially for TNF-α (Fig 13, p<0.05) and IL-6 (Fig 14, p<0.05). The remaining four species had lower proinflammatory activity either for LPGs or GIPLs (Figs 13–16, P<0.05). IL-12 production was very low for all *Viannia* glycoconjugates (Fig 15, P<0.05). All data are summarized in Table 3.

## Type II LPG in *L.* (*V.*) *lainsoni* had higher pro-inflammatory activity in murine macrophages

To investigate whether the variations in the functional properties of glycoconjugates were due to biochemical polymorphisms, they were depolymerized for biochemical analysis (Fig 17). Lipophosphoglycan repeat units were purified and subjected to FACE analysis. Most of repeat units co-migrated with oligosaccharide standard $G_2$ and type I control from *L.* (*L.*) *infantum*

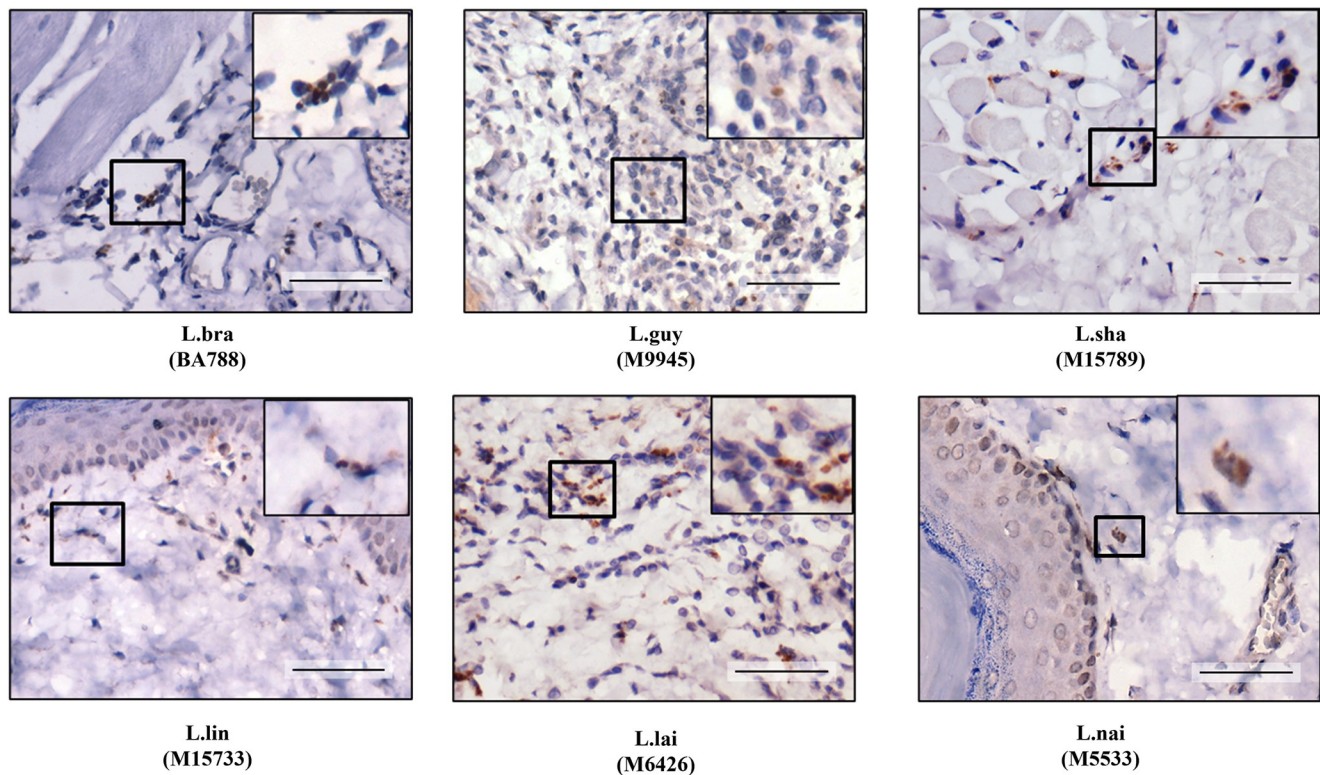

**Fig 9. *Leishmania* (*Viannia*) species had different parasite load in the dermis of *M. auratus* 20 days PI by immunohistochemistry (IHC).** Upper sections in higher magnification showing amastigote forms. Legend: L.braz, *L. braziliensis*; L.guy, *L. guyanensis*; L.sha, *L. shawi*, L.lind, *L. linderbergi*, L.lai, *L. lainsoni*, L.nai, *L. naiffi*. Scale bar = 20 um.

(BA262) confirming the Gal(β1,4)Man structure common to all LPGs (Fig 17A and 17B). All species, except *L. (V.) lainsoni*, had LPG devoid of sidechains (type I) (Fig 17A and 17B). *L. (V.) lainsoni* had one side-chain co-migrating close to oligo-glucose ladder $G_3$, suggestive of a hexose sidechain (Type II LPG) (Fig 17B, dark arrow). Like LPGs, most GIPLs structures were not polymorphic (Fig 17C and 17D). Most GIPLs were rich in galactose (type II) and only *L. (V.) guyanensis* was rich in galactose/mannose (type I or hybrid) (Fig 17C, dark arrow). All data are summarized in Table 3.

### *Leishmania* (V*iannia*) *lainsoni* clustered differently in comparison to other *L. (Viannia)* species

The biological, biochemical, and immunological parameters studied in this paper were collected and summarized in Tables 2 and 3. To provide a better understanding of the taxonomic relationships among the *L. (Viannia)* species, Principal Component Analysis (PCA) was performed, and we found that they clustered into two groups (Fig 18A). One was composed by *L. (V.) lainsoni* (cluster 1) and another by the five remaining species (cluster 2). After hierarchical cluster by squared-Euclidean distance, cluster 2 was splitted into two subdivisions: *L. (V.) lindenbergi/L. (V.) naiffi* and *L. (V.) guyanensis/L. (V.) shawi/L. (V.) braziliensis* (Fig 18B).

## Discussion

Several factors affect infectivity and/or pathogenicity amongst *Leishmania* strains/species. In our search for a model to study those species, we used different *L. (Viannia)* strains evaluating

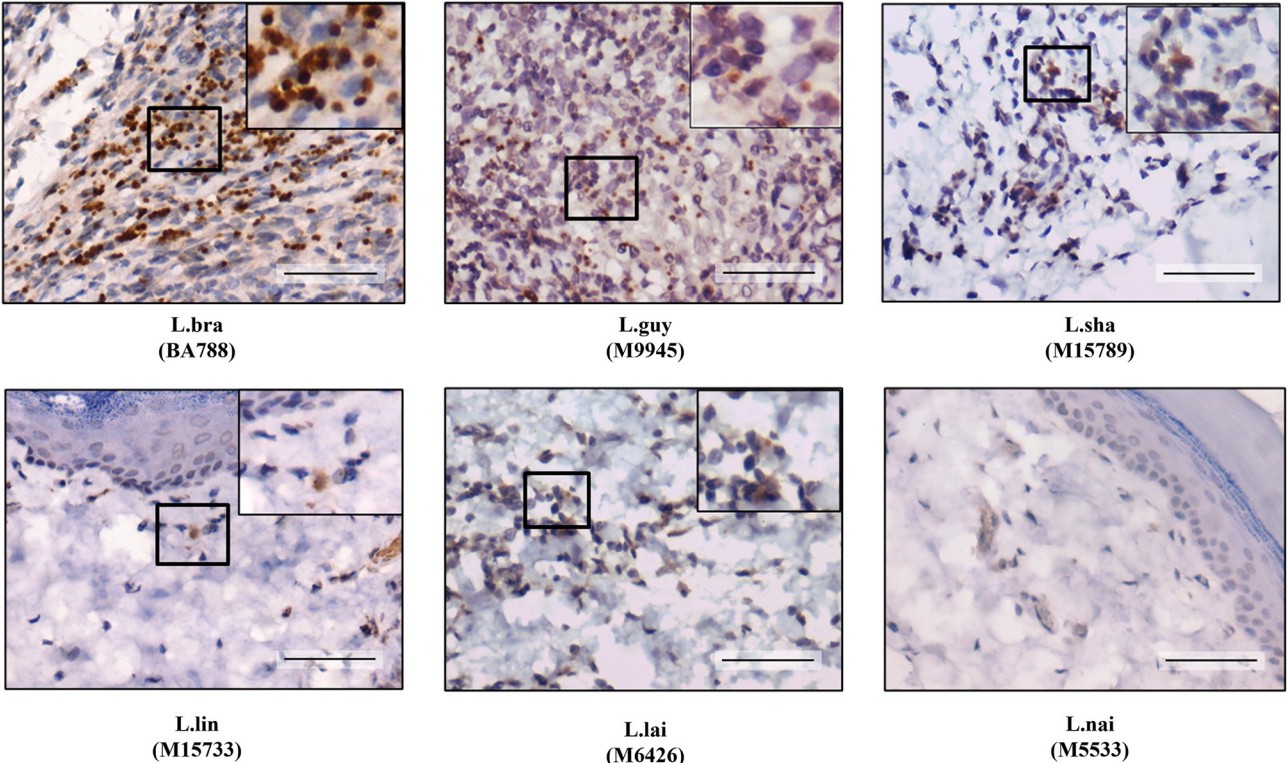

**Fig 10. Imunohistochemistry (IHC) of *Leishmania* (*Viannia*) species in the dermis of *M. auratus* after 40 days PI shows clearance of *L. naiffi* parasites.** Upper sections in higher magnification showing amastigote forms. Legend: L.braz, *L. braziliensis*; L.guy, *L. guyanensis*; L.sha, *L. shawi*, L.lind, *L. linderbergi*, L.lai, *L. lainsoni*, L.nai, *L. naiffi*. Scale bar = 20 µm.

their ability to infect two types of macrophages *in vitro* including resident peritoneal (hamster) and continuous cell line THP-1 (human). Previous studies with other species/strains of *Leishmania* demonstrated that *L. (L.) infantum* (dermotropic/viscerotropic), *L. (L.) amazonensis* and *L. (V.) braziliensis* were able to successfully infect hamsters and THP-1 macrophages [32,39,40]. Here, all species were able to infect both cell types displaying variability in their intracellular parasitism. Resident peritoneal macrophages were more susceptible to infection than THP-1 cells exhibiting higher parasitism depending on the *Leishmania* species. In resident peritoneal macrophages, *L. (V.) guyanensis* and *L. (V.) shawi* were the most infective. *L. (V.) naiffi* and *L. (V.) lindenbergi* were the least infective species for both cell types. Regarding intracellular development, *L. (V.) guyanensis* had a higher amastigote number (~2-3-fold) than the other species. These data advocate the use of resident peritoneal macrophages from hamsters instead of THP-1 cells. The two less infective species, *L. (V.) naiffi* and *L. (V.) lindenbergi* still warrant a better model of cellular infection. In nature, *L. (V.) guyanensis* strains are often infected with the *Leishmania RNA virus*, whereas *L. (V.) braziliensis* and *L. (L.) infantum* are not [46,47,49–51]. The literature suggests a correlation between LRV1 presence and severity of disease leading to mucosal development for humans [16]. We found this virus in *L. (V.) guyanensis* and *L. (V.) shawi*. The former had a higher viral load and intracellular infection than the latter. Although it is tempting to suggest that LRV1 presence may increase infectivity/intracellular parasitism, at least for *L. (V.) guyanensis*, the present study did not find a coherent correlation between LRV1 presence and invasion/intracellular replication in our *in vitro* models. Our observations that these *L. (Viannia)* species developed more efficiently in hamster cells, led us to use *M. auratus* for our *in vivo* experiments.

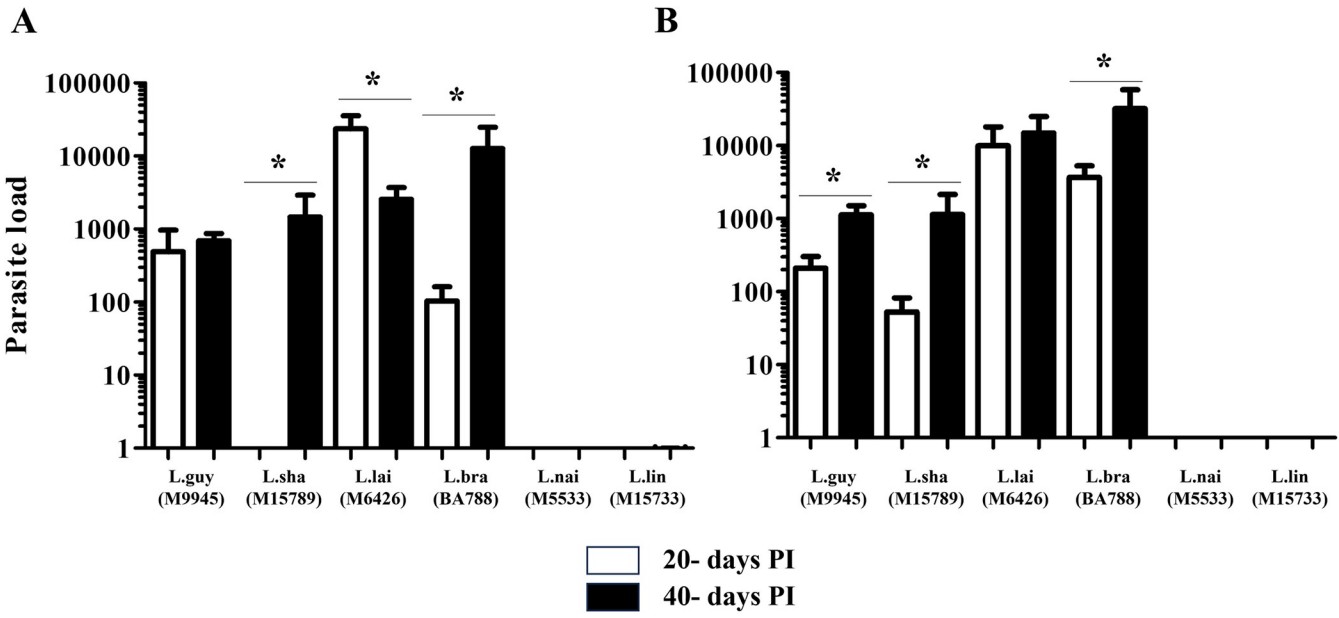

**Fig 11. *Leishmania naiffi* and *L. lindenbergi* had the lowest parasite load in *M. auratus* tissues.** Parasite load (parasites/mg tissue) recovered using the limiting dilution method from the dermis (A) and lymph nodes (B) of *M. auratus* infected with different *Leishmania*. Legend: L.braz, *L. braziliensis*; L.guy, *L. guyanensis*; L.sha, *L. shawi*, L.lind, *L. linderbergi*, L.lai, *L. lainsoni*, L.nai, *L. naiffi*. White and dark bars indicate 20- and 40- days PI, respectively. Asterisks indicate statistical differences (P<0.05).

The novelty of our work was to use for the first time SGE from *Ny. neivai* since saliva compounds have several immunomodulatory activities [43,52–54]. All strains had low ability to develop macroscopic ulcerated lesions in hamsters´ footpads. The only species that caused external visible swelling was *L.* (*V.*) *guyanensis* that increased over time. Like *in vitro* experiments, in our experimental *in vivo* conditions, LRV1 could not solely explain variations in pathology among species/strains, since LRV1-positive *L.* (*V.*) *shawi* did not cause external lesion either. However, while analyzing the microscopic dermis of infected footpads this picture changed. At both PI intervals, very productive inflammatory reactions in the dermis for some *L.* (*Viannia*) species were noticed by HE and IHC. We observed a strong inflammatory exudate for at least four species including *L.* (*V.*) *braziliensis*, *L.* (*V.*) *guyanensis*, *L.* (*V.*) *shawi* and *L.* (*V.*) *lainsoni*. Those species were even recovered after limiting dilution method from skin and draining lymph nodes. No inflammatory infiltration was seen in *L.* (*V.*) *naiffi*/*L.* (*V.*) *lindenbergi* after 40 days PI suggesting clearance of those parasites. Altogether, that data confirmed that for at least four *L.* (*Viannia*) species were able to trigger important immunopathological responses in the dermis of *M. auratus* in the absence of macroscopical external lesions. A distinguished feature of *L.* (*V.*) *braziliensis* strain used here was its ability to visceralize. It is already known that depending on the inoculum, that dermotropic species possess different abilities to colonize animal viscera. Several reports already demonstrate this phenomenon for other strains of *L.* (*V.*) *braziliensis* [33], *L.* (*M.*) *enriettii* [55], *L.* (*L.*) *amazonensis* [56] and *L.* (*L.*) *infantum* [45]. Here, we have shown that although most *L.* (*Viannia*) strains had variable levels of pathology in the dermis, only *L.* (*V.*) *braziliensis* visceralized to the liver evoking two types of lesions with presence of giant cells, lymphocytes, and fibroblasts. Consistent with previous observations, in the liver, a certain degree of infection control is detected due to the development of granulomas, this may justify the complete absence of parasites in those lesions [32,57,58]. However, no visible dermotropic lesions appeared in this species during the time

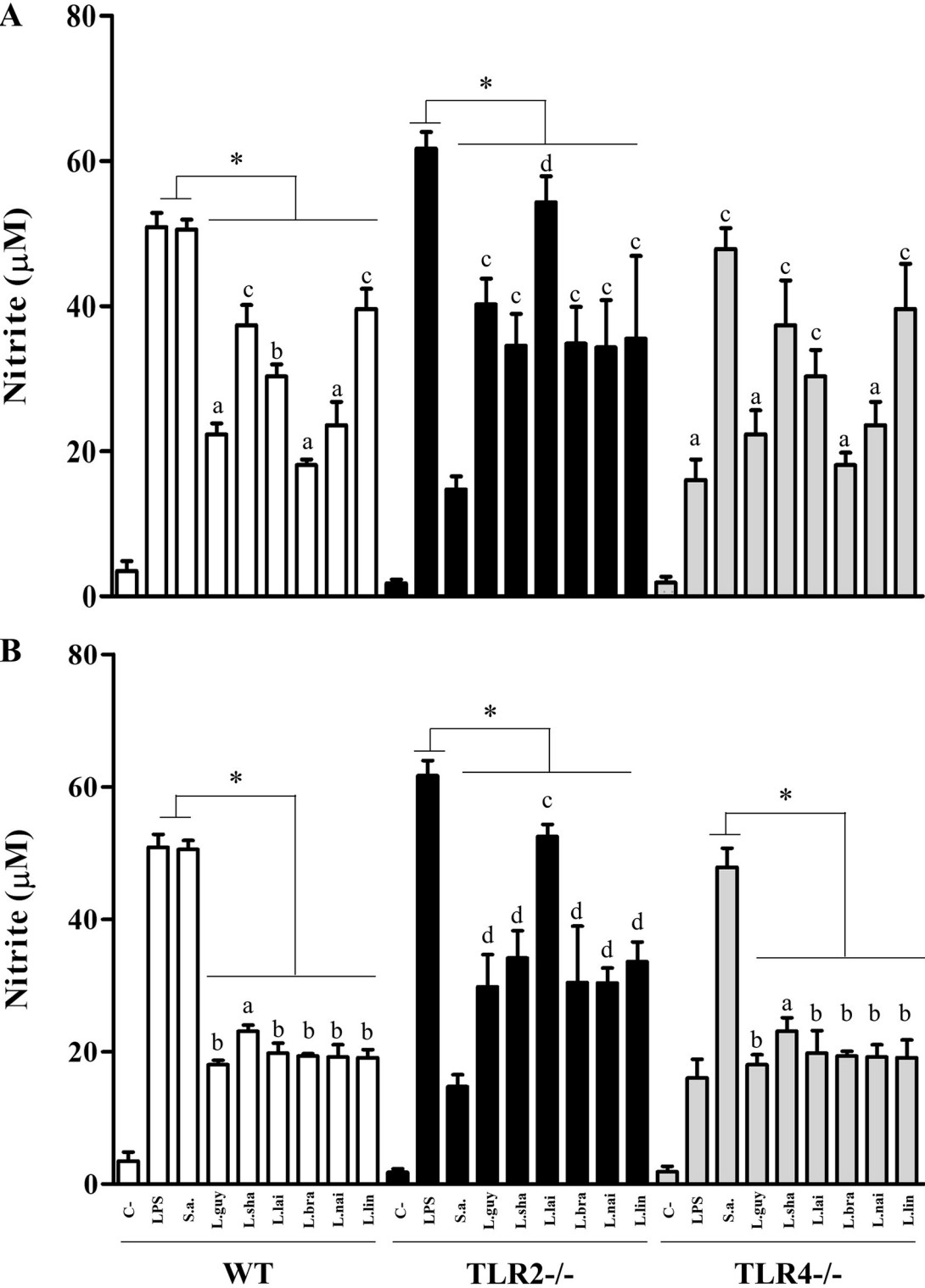

**Fig 12. Glycoconjugates (LPGs and GIPLs) from *Leishmania* (*Viannia*) species induced variable levels of Nitric oxide (µM) by C57BL/6 murine macrophages *via* TLR4/TLR2.** Wild-type (WT, white bars), TLR2 (-/-) (dark bars) and TLR4 (-/-) (gray bars) knockouts. Macrophages were exposed to lipophosphoglycans (LPGs) (A) and glycoinositolphospholipids (GIPLs) (B) (10 µg/mL) from different *Leishmania*. Legend: C-, negative control (RPMI medium); LPS, lipopolysaccharide (TLR4 agonist, 100 ng/mL); S.a., *Staphylococcus aureus* extract (TLR2 agonist, 100 ng/mL); L.braz, *L. braziliensis*; L.guy, *L. guyanensis*; L.sha, *L. shawi*, L.lind, *L. linderbergi*, L.lai, *L. lainsoni*, L.nai, *L. naiffi*. Asterisks indicate statistical differences within the same mice lineage (P<0.05). The same letters above bars do not indicate statistical differences (P>0.05). Different letters indicate statistical differences (P<0.05).

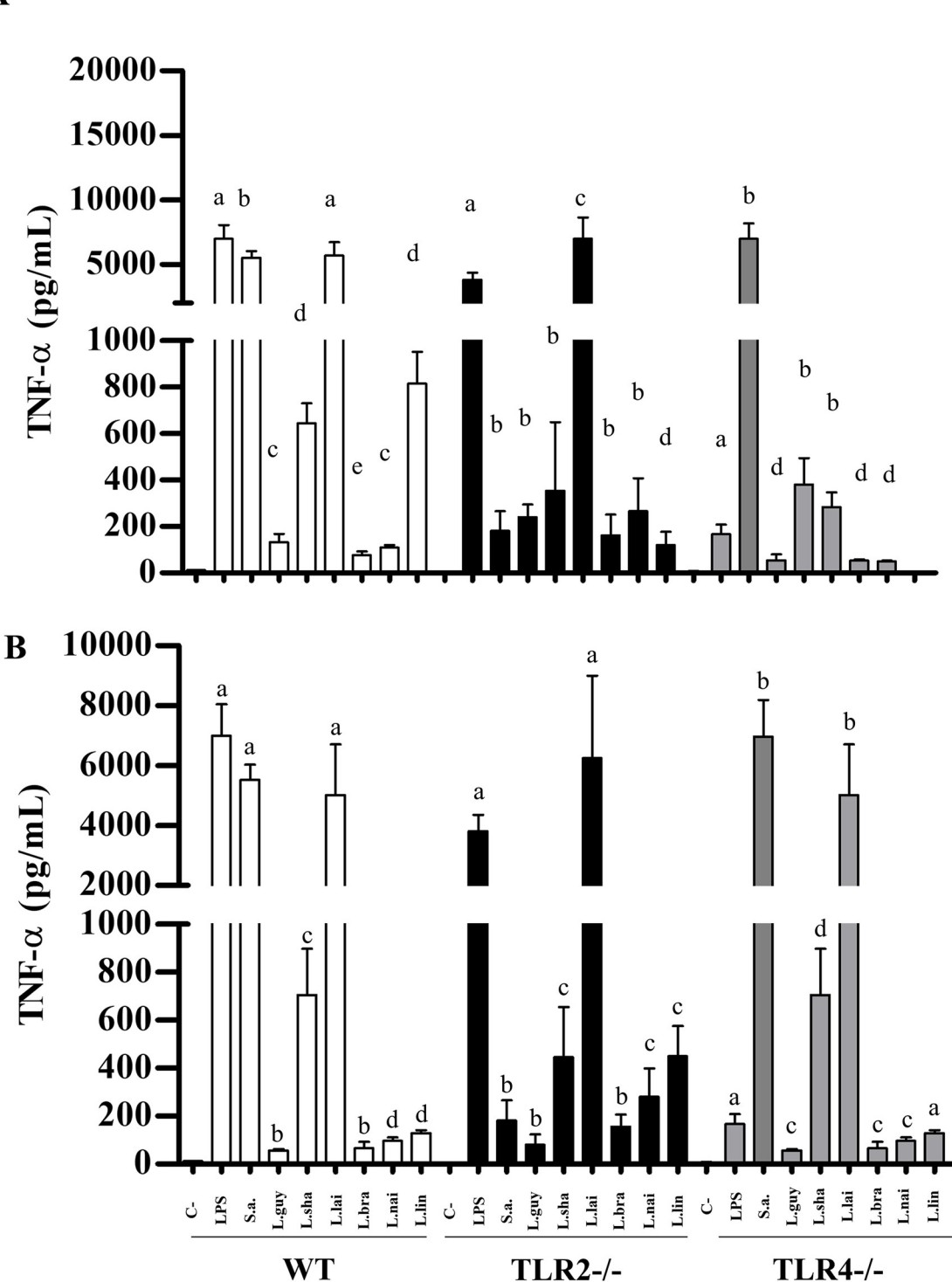

**Fig 13. *Leishmania lainsoni* glycoconjugates induced higher TNF-α production by C57BL/6 murine macrophages *via* TLR4/TLR2.** Wild-type (WT, white bars), TLR2 (-/-) (dark bars) and TLR4 (-/-) (gray bars) knockouts. Macrophages were exposed to lipophosphoglycans (LPGs) (A) and glycoinositolphospholipids (GIPLs) (B) (10 μg/mL) from different *Leishmania*. Legend: C-, negative control (RPMI medium); LPS, lipopolysaccharide (TLR4 agonist, 100 ng/mL); S.a., *Staphylococcus aureus* extract (TLR2 agonist, 100 ng/mL); L.braz, *L. braziliensis*; L.guy, *L. guyanensis*; L.sha, *L. shawi*, L.lind, *L. linderbergi*, L.lai, *L. lainsoni*, L.nai, *L. naiffi*. Asterisks indicate statistical differences within the same mice lineage (P<0.05). The same letters above bars do not indicate statistical differences (P>0.05). Different letters indicate statistical differences (P<0.05).

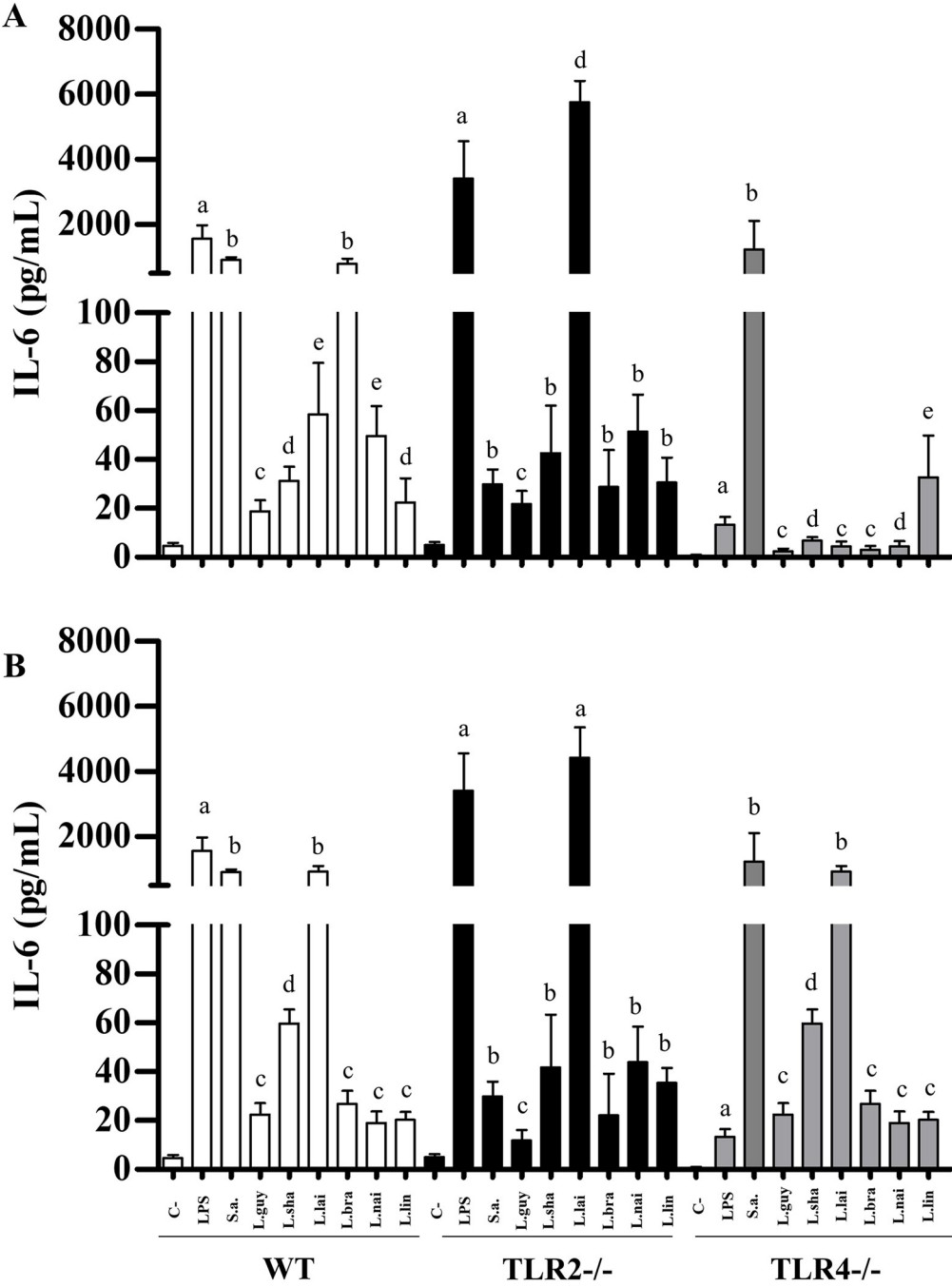

**Fig 14.** *Leishmania lainsoni* **glycoconjugates induced higher IL-6 production by C57BL/6 murine macrophages** *via* **TLR4/TLR2.** Wild-type (WT, white bars), TLR2 (-/-) (dark bars) and TLR4 (-/-) (gray bars) knockouts. Macrophages were exposed to lipophosphoglycans (LPGs) (A) and glycoinositolphospholipids (GIPLs) (B) (10 μg/mL) from different *Leishmania*. Legend: C-, negative control (RPMI medium); LPS, lipopolysaccharide (TLR4 agonist, 100 ng/mL); S.a., *Staphylococcus aureus* extract (TLR2 agonist, 100 ng/mL); L.braz, *L. braziliensis*; L.guy, *L. guyanensis*; L. sha, *L. shawi*, L.lind, *L. linderbergi*, L.lai, *L. lainsoni*, L.nai, *L. naiffi*. Asterisks indicate statistical differences within the same mice lineage (P<0.05). The same letters above bars do not indicate statistical differences (P>0.05). Different letters indicate statistical differences (P<0.05).

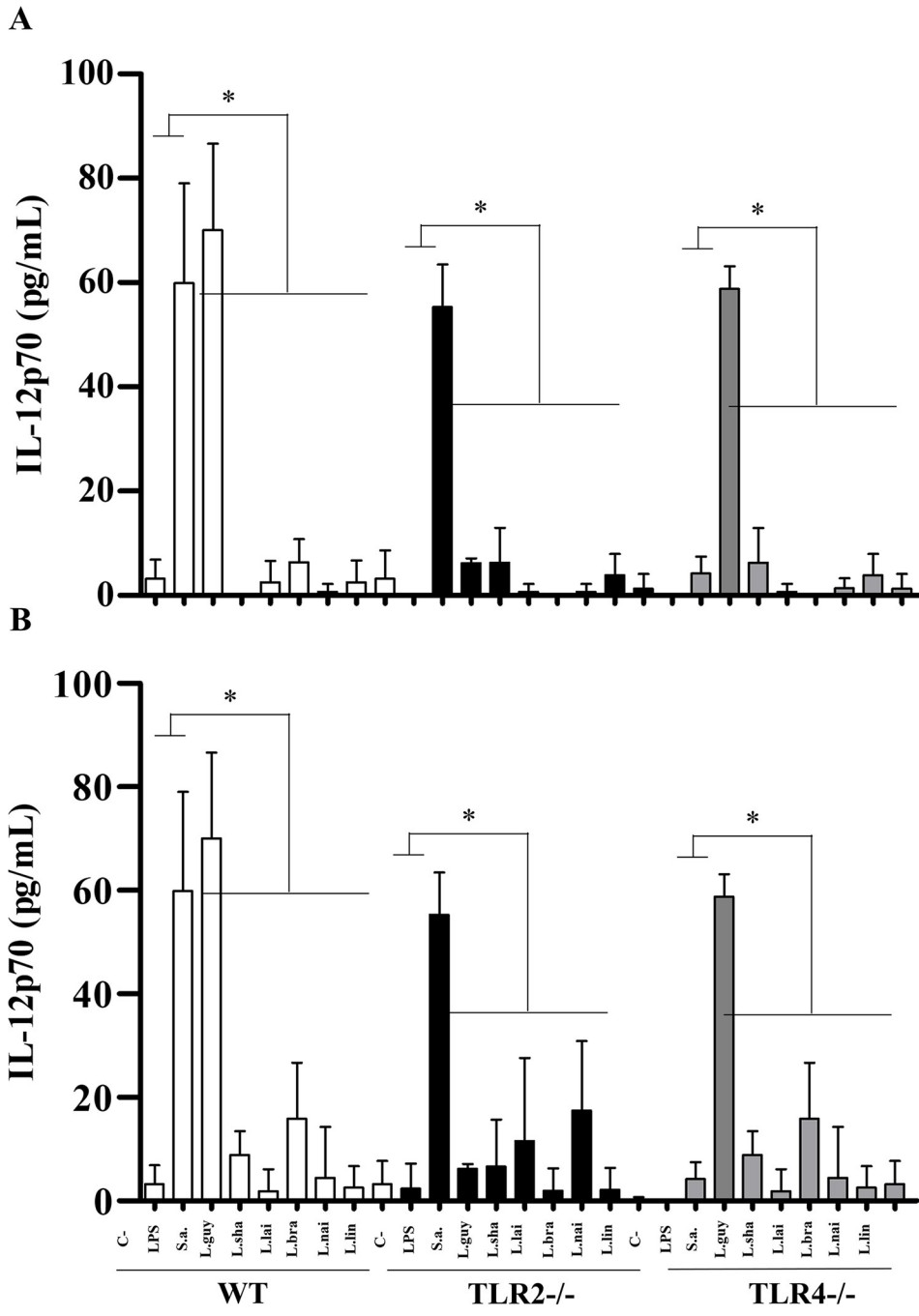

**Fig 15. *Leishmania* (*Viannia*) species did not induce considerable levels of IL-12 by C57BL/6 murine macrophages.** Wild-type (WT, white bars), TLR2 (-/-) (dark bars) and TLR4 (-/-) (gray bars) knockouts. Macrophages were exposed to lipophosphoglycans (LPGs) (A) and glycoinositolphospholipids (GIPLs) (B) (10 μg/mL) from different *Leishmania*. Legend: C-, negative control (RPMI medium); LPS, lipopolysaccharide (TLR4 agonist, 100 ng/mL); S.a., *Staphylococcus aureus* extract (TLR2 agonist, 100 ng/mL); L.braz, *L. braziliensis*; L.guy, *L. guyanensis*; L.sha, *L. shawi*, L.lind, *L. linderbergi*, L.lai, *L. lainsoni*, L.nai, *L. naiffi*. Asterisks indicate statistical differences within the same mice lineage (P<0.05).

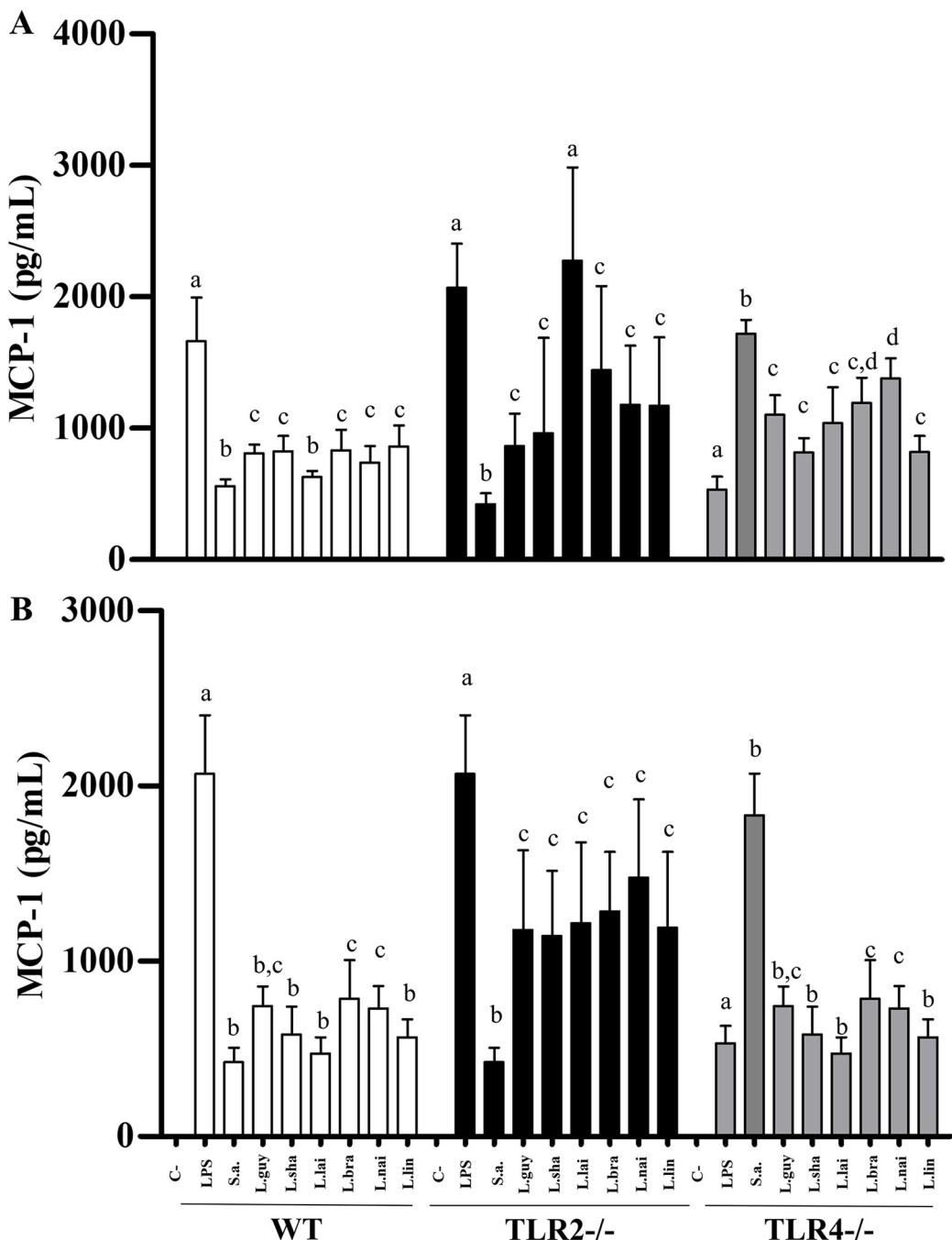

**Fig 16. MCP-1 production by C57BL/6 murine macrophages differ in LPG-stimulated but not in GIPLs-stimulated from *Leishmania* (*Viannia*) species.** Wild-type (WT, white bars), TLR2 (-/-) (dark bars) and TLR4 (-/-) (gray bars) knockouts. Macrophages were exposed to lipophosphoglycans (LPGs) (A) and glycoinositolphospholipids (GIPLs) (B) (10 µg/mL) from different *Leishmania*. Legend: C-, negative control (RPMI medium); LPS, lipopolysaccharide (TLR4 agonist, 100 ng/mL); S.a., *Staphylococcus aureus* extract (TLR2 agonist, 100 ng/mL); L.braz, *L. braziliensis*; L.guy, *L. guyanensis*; L.sha, *L. shawi*, L.lind, *L. linderbergi*, L.lai, *L. lainsoni*, L.nai, *L. naiffi*. Asterisks indicate statistical differences within the same mice lineage (P<0.05). The same letters above bars do not indicate statistical differences (P>0.05). Different letters indicate statistical differences (P<0.05).

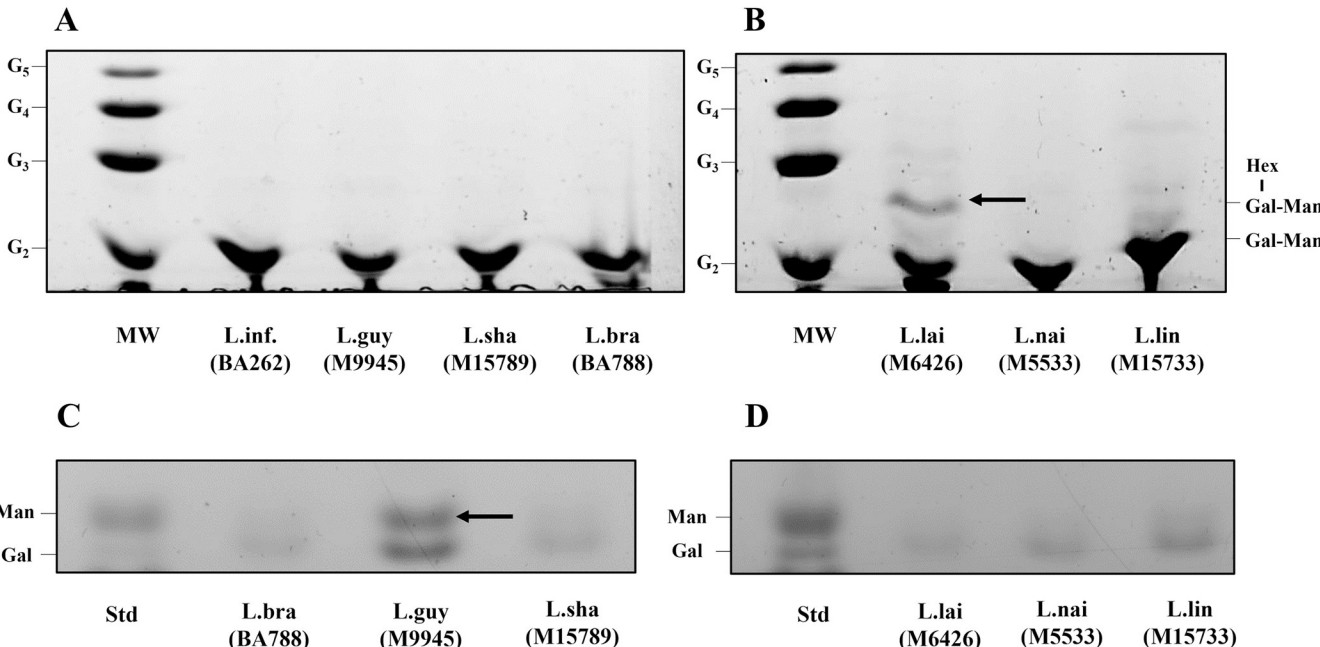

**Fig 17. Glycoconjugates of *Leishmania* (*Viannia*) species display interspecies polymorphisms in their carbohydrate's motifs.** Fluorophore-assisted carbohydrate electrophoresis (FACE) analysis of depolymerized glycoconjugates (lipophosphoglycans and glycoinositolphospholipids) from *Leishmania*. A) and B), polysaccharide profile of dephosphorylated repeat units of LPGs; C) and D), monosaccharide profile of GIPLs. Legend: L.inf, *L. infantum* (type I LPG —control), L.braz, *L. braziliensis*; L.guy, *L. guyanensis*; L.sha, *L. shawi*; L.lind, *L. linderbergi*; L.lai, *L. lainsoni*; L.nai, *L. naiffi*; MW, molecular weight represented by oligoglucose ladder $G_2$-$G_5$; Std, monosaccharide standard represented by galactose and mannose; Gal, galactose; Man, Mannose and Hex– Hexose.

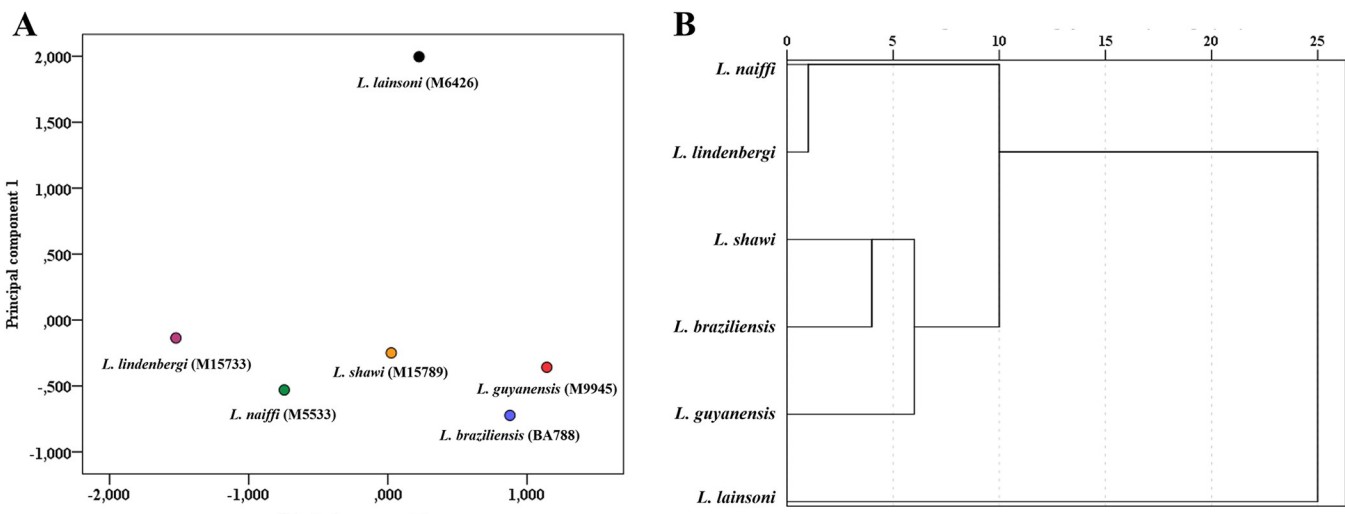

**Fig 18. *Leishmania lainsoni* is clustered separately from other members within the *Viannia* subgenus.** Principal component analysis (PCA) derived from 33 variables (Tables 2 and 3). A) Dotplot graphic showing distinct biological patterns among the different *Viannia* species. B) Hierarchical cluster by squared-Euclidean distance using the first two PCs indicating the proximity of the six *Viannia* species and the formation of an isolated cluster composed of *L. lainsoni*.

proposed (up to 7 weeks PI). We therefore suggest that for this strain (BA788) a longer period of 8–10 weeks should be performed to circumvent the possible lack of apparent lesions.

Like *in vitro* data, *in vivo* assays confirmed that all six *L.* (*Viannia*) species varied in their ability to cause infection in hamsters. *L.* (*V.*) *lindenbergi* and *L.* (*V.*) *naiffi*) had low ability to sustain infection in those animals and there is still the need to find alternative hosts. Regarding the other species, except for *L.* (*V.*) *guyanensis*, that caused apparent macroscopic swelling, the hamster model showed that parasites were microscopically viable even at 40-days PI. For those remaining four species, the hamster model could provide information on inflammatory events, cellular attraction and even visceralization. Since variable levels of infectivity and proinflammatory activity were reported in the previous findings, our next step was to investigate how parasite virulence factors could modulate such events.

*Leishmania* species possess a complex array of surface molecules acting as Pathogen-Associated Molecular Patterns (PAMPs). They include glycoprotein 63 (GP-63), lipophosphoglycans (LPGs) and glycoinositolphospholipids (GIPLs) [59–61]. TLR2 and TLR4 are the main receptors recognized by LPGs and GIPLs and the ability to induce proinflammatory responses by several species was dependent on the species strains [37,38,62–65]. Here, glycoconjugates from all *L.* (*Viannia*) species demonstrated the ability to trigger cytokine/chemokine and NO production like other dermotropic species strains of *Leishmania*. Confirming previous data with New World *L.* (*Viannia*) and *L.* (*Leishmania*) species that *L.* (*Viannia*) LPGs were also able to activate TLR4 and secondarily TLR2 [37,40,65]. Interestingly, this activation pattern was more homogeneous for the GIPLs, that were primarily TLR4 agonists like those from *L.* (*L.*) *infantum*/*L.* (*V.*) *braziliensis* and *Trypanosoma cruzi* [62,66]. However, both glycoconjugates were not potent IL-12 inducers confirming the role of this important cytokine during *Leishmania* interaction with the host. The low ability of *Leishmania* glycoconjugates in inducing this cytokine may suggest an escape mechanism of those parasites during early events in the innate immune compartment [67]. For the other mediators, in general, a higher agonistic activity was observed for *L.* (*V.*) *lainsoni* and sometimes for *L.* (*V.*) *shawi*/*L.* (*V.*) *lindenbergi* glycoconjugates, especially for NO and TNF-α. The variable potential of *L.* (*Viannia*) glycoconjugates corroborates the immunopathology caused by these dermotropic species. They often cause ulcerated lesions in the patient's skin and some of them can even lead to mucosal involvement [16,68]. This is different from the dermotropic *L.* (*L.*) *amazonensis* LPG that can trigger strong proinflammatory responses but does not translocate NF-kB [64]. Here, even though we could not observe productive macroscopical lesions in the hamsters' skin, we detected microscopical exudates and inflammatory infiltration suggesting the involvement of macrophages and several mediators during infection. However, our next step was to biochemically characterize those glycoconjugates to establish a possible functional relationship between structure and activity.

LPG and GIPLs showed variations in structures of *L.* (*V.*) *lainsoni* LPG and *L.* (*V.*) *guyanensis* GIPLs. Of all *L.* (*Viannia*) species reported here, only *L.* (*V.*) *braziliensis* had its LPG/GIPL characterized. LPGs from this species are from three types (I-III) depending on the sidechains bearing the conserved Gal(β1,4) Man(α1)-PO$_4$ repeat unit motif [37,69]. The GIPLs from *L.* (*V.*) *braziliensis* are mostly galactosylated and considered type II [62]. Here, we provide for the first time the structures of the repeat units of other *L.* (*Viannia*) species. Most of the LPGs are devoid of side chains (type I), except for that of *L.* (*V.*) *lainsoni* that showed a polymorphism having a hexose sidechain (type II). Type I LPG is the most common and reported for in several species/strains of *Leishmania* [37,38,62,64,69]. The *L.* (*V.*) *lainsoni* LPG is very similar to those reported for *L.* (*L.*) *mexicana* [70] and *L.* (*L.*) *infantum* (strain PP75) [35], who also possess a hexose (glucose) as sidechains. With respect to GIPLs, except for *L.* (*V.*) *guyanensis*, most of them were galactosylated (Type II). A prominent mannose band detected by FACE

analysis in addition to the galactose band is suggestive of a Type I or hybrid GIPL commonly observed previously for *L.* (*L.*) *infantum/L.* (*L.*) *donovani* [62] and *L.* (*L.*) *tropica/L.* (*L.*) *aethiopica* [71]. Overall, the present data indicates that the polymorphisms detected in the carbohydrate motifs were very low. Except for *L.* (*V.*) *lainsoni* LPG whose polymorphism in the carbohydrate motif resulted in higher proinflammatory activity, this was not very clear for *L.* (*V.*) *guyanensis* type I/hybrid GIPL. This is in accordance with previous literature on glycobiology suggesting that other parts of the glycoconjugates, e.g. lipids, also have proinflammatory activity [62,72]. However, it is very clear that *L.* (*Viannia*) glycoconjugates have a pivotal role during early events of *Leishmania* infection in the innate immune compartment.

The present multidisciplinary study used several biological, biochemical, and immunological parameters to elucidate the parasite/mammalian-host interface of the less common *L.* (*Viannia*) species. All data were collected, summarized, and subjected to a PCA analysis to understand possible taxonomical relationships among *L.* (*Viannia*) strains. Confirming previous questions in literature [24,73] the PCA analysis, based on the parameters evaluated in the present study, showed that *L.* (*V.*) *lainsoni* is separated from the other *L.* (*Viannia*) species. We conclude that phenotypic and intrinsic characters show that *L.* (*V.*) *lainsoni* belongs to the viannian clade but is distinct from all other *L.* (*Viannia*) species. This opens the possibility for WGS analysis for a better understanding of its taxonomic position within the *L.* (*Viannia*) subgenus.

In conclusion, at least four *L.* (*Viannia*) species display polymorphisms in their abilities to infect and sustain infection in *M. auratus* upon needle injection in the presence of sandfly SGE. This advocates the use of this model for four *L.* (*Viannia*) species specially to understand their pathological mechanisms in the dermis. Their glycoconjugates display low polymorphisms but possess high proinflammatory agonistic activity *via* TLR4/TLR2. The establishment of a good animal model would be important not only to understand host-parasite relationships but also to provide *in vivo* testing of new drugs for chemotherapy. Besides, the ability of the glycoconjugates from all *Leishmania Viannia* species in inducing proinflammatory responses in macrophages suggest their possible use as vaccines adjuvants.

## Supporting information

**S1 Fig. *Leishmania* (*Viannia*) species display lower levels of infection in THP-1 macrophages and lack correlation with LRV1 presence.** (A) Macrophage infection (%), (B) number of amastigotes per macrophage and (C) Intracellular amastigote forms in THP-1 macrophages. Legend: L.braz, *L. braziliensis*; L.guy, *L. guyanensis*; L.sha, *L. shawi*, L.lind, *L. linderbergi*, L.lai, *L. lainsoni* and L.nai, *L. naiffi*. Magnification of 1000x. Dark and white bars indicate LRV1 presence and absence, respectively. Letters above bars indicate statistical differences (P<0.05). (TIF)

**S2 Fig. Representative negative controls of skin (A), liver (B) and spleen (C) of uninfected *M. auratus* stained with hematoxylin-eosin (HE).** Scale bar = 20 μm. (TIF)

**S1 File. Raw data and descriptive statistics.** (DOCX)

## Acknowledgments

We thank the Flow Cytometry and Sequencing Platform of René Rachou Institute (IRR/FIO-CRUZ) for performing CBA analysis and sequencing, respectively. We thank the Animal

Facility of the Faculty of Medicine (FMUSP) for helping with *in vivo* assays using *M. auratus*. We thank the Animal Facility of IRR/FIOCRUZ for providing mice for macrophage assays.

## Author Contributions

**Conceptualization:** Rodrigo Pedro Soares, Márcia Dalastra Laurenti.

**Data curation:** Felipe Dutra-Rêgo, Gabriela Venícia Araujo Flores, Carmen Maria Sandoval Pacheco.

**Formal analysis:** Rodrigo Pedro Soares, Felipe Dutra-Rêgo, Gabriela Venícia Araujo Flores, Carmen Maria Sandoval Pacheco, Jeffrey Jon Shaw.

**Funding acquisition:** Rodrigo Pedro Soares, Márcia Dalastra Laurenti.

**Investigation:** Igor Campos Fontes, Felipe Dutra-Rêgo, Jeronimo Nunes Rugani, Paulo Otávio L. Moreira, Vânia Lúcia Ribeiro da Matta, Gabriela Venícia Araujo Flores, Carmen Maria Sandoval Pacheco.

**Methodology:** Jeronimo Nunes Rugani, Paulo Otávio L. Moreira, Gabriela Venícia Araujo Flores, Carmen Maria Sandoval Pacheco, Andrey José de Andrade, Magda Clara Vieira da Costa-Ribeiro.

**Project administration:** Rodrigo Pedro Soares, Márcia Dalastra Laurenti.

**Resources:** Rodrigo Pedro Soares, Andrey José de Andrade, Magda Clara Vieira da Costa-Ribeiro, Jeffrey Jon Shaw, Márcia Dalastra Laurenti.

**Supervision:** Rodrigo Pedro Soares, Márcia Dalastra Laurenti.

**Validation:** Rodrigo Pedro Soares, Felipe Dutra-Rêgo, Paulo Otávio L. Moreira.

**Visualization:** Igor Campos Fontes, Felipe Dutra-Rêgo, Vânia Lúcia Ribeiro da Matta, Gabriela Venícia Araujo Flores, Carmen Maria Sandoval Pacheco, Jeffrey Jon Shaw.

**Writing – original draft:** Rodrigo Pedro Soares.

**Writing – review & editing:** Rodrigo Pedro Soares, Igor Campos Fontes, Felipe Dutra-Rêgo, Jeronimo Nunes Rugani, Paulo Otávio L. Moreira, Vânia Lúcia Ribeiro da Matta, Gabriela Venícia Araujo Flores, Carmen Maria Sandoval Pacheco, Andrey José de Andrade, Magda Clara Vieira da Costa-Ribeiro, Jeffrey Jon Shaw, Márcia Dalastra Laurenti.

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
