## [Decision Letter · Decision Letter 0]

17 Jun 2024

Dear Mr. Soares,

Thank you very much for submitting your manuscript "Unveiling the Enigmatic nature of Amazonian Leishmania (Viannia) species: their Leishmania RNA viruses, Virulence, and Histopathology" for consideration at PLOS Neglected Tropical Diseases. As with all papers reviewed by the journal, your manuscript was reviewed by members of the editorial board and by several independent reviewers. In light of the reviews (below this email), we would like to invite the resubmission of a significantly-revised version that takes into account the reviewers' comments. 

We cannot make any decision about publication until we have seen the revised manuscript and your response to the reviewers' comments. Your revised manuscript is also likely to be sent to reviewers for further evaluation.

Sincerely,

Ulisses Gazos Lopes

Academic Editor

Susan Madison-Antenucci

Section Editor

Reviewer's Responses to Questions

**Key Review Criteria Required for Acceptance?**

**Methods**

-Are the objectives of the study clearly articulated with a clear testable hypothesis stated?

-Is the study design appropriate to address the stated objectives?

-Is the population clearly described and appropriate for the hypothesis being tested?

-Is the sample size sufficient to ensure adequate power to address the hypothesis being tested?

-Were correct statistical analysis used to support conclusions?

-Are there concerns about ethical or regulatory requirements being met?

Reviewer #1: The methods are well defined with few issues. Please verify general comments.

Reviewer #2: (No Response)

Reviewer #3: The objectives of the study are unclear in the introduction, but they were specified in the abstract and in the author's summary, and are articulated with a testable hypothesis. The study design is appropriate to address the stated objectives and the population is appropriate for the hypothesis tested, with the sample size being sufficient to ensure adequate power to address the hypothesis. Statistical analysis is adequate and ethical requirements were met.

Line 185–187 - Were the negative and positive controls placed after treating the macrophages with INF-gamma? If yes, this should be clear.

**Results**

-Does the analysis presented match the analysis plan?

-Are the results clearly and completely presented?

-Are the figures (Tables, Images) of sufficient quality for clarity?

Reviewer #1: Please see general comments.

Reviewer #2: (No Response)

Reviewer #3: The results are presented clearly and completely, with figures and tables of sufficient quality to understand the data. However, some points deserve attention:

Table 2 - Please, put the species and not just the strains in the first column.

Figures 2, 12-16 - The relationship between the letters and the P value is unclear. Please describe this relationship in each subtitle.

Figure 3 - To know when the lesion began to be significant, it would be interesting to compare the size of the paw edema on the days of infection with the size of the paw before infection. Was this comparison carried out? If yes, indicate in the figure and clarify in the subtitle.

The hamster is known to be a good experimental model used for L. braziliensis. However, according to our experience in the laboratory, the lesion usually becomes significant around the sixth/seventh week after infection. It is possible that 5 weeks of infection (40 days PI) was not enough time for the lesion to be macroscopically visible to other species, since not even L. braziliensis the lesion became visible. This subject must be present in the discussion.

Figure 13A - Why does the scale continue up to 100,000 if the values end at 20,000? Consider finishing at 20,000.

**Conclusions**

-Are the conclusions supported by the data presented?

-Are the limitations of analysis clearly described?

-Do the authors discuss how these data can be helpful to advance our understanding of the topic under study?

-Is public health relevance addressed?

Reviewer #1: The conclusions need be updated. Please verify general comments.

Reviewer #2: (No Response)

Reviewer #3: The conclusions are supported by the data presented, and the limitations of the analysis have been described. The authors discuss how these data can be useful in advancing our understanding of the topic under study, and also address the relevance of the study to public health. However, some points could be better explored, such as, for example, the importance of establishing experimental models for the Leishmania species covered in the study.

Establishing an experimental model that allows us to unravel the mechanisms of the complex host-parasite relationship is essential. This knowledge can serve as a basis for, for example, developing new, more effective, and less toxic, chemotherapy drugs for leishmaniasis treatment. This could be one of the biases to be explored in the introduction, since the working points to immune mechanisms present in the parasite-host relationship, which could serve, in subsequent studies, as possible targets for immunomodulators to treat these parasitoses.

**Editorial and Data Presentation Modifications?**

Reviewer #1: All figures need be revised. See general comments.

Reviewer #2: (No Response)

Reviewer #3: (No Response)

**Summary and General Comments**

Reviewer #1: Dear Authors,

The rationale of your manuscript is well-presented, but the manuscript requires several improvements, particularly concerning the lengthy format which needs to be updated to be more concise. I recommend rewriting the manuscript with a clearer, more focused aim. It would be beneficial to first present the analysis of parasites characteristics followed by the infection patterns. Below are some specific considerations:

Abstract: The abstract contains double the words recommended by PNTD. Please reduce it to the essential content.

Introduction: While the Introduction is clear, it is overly extensive. Please include only the essential background to support the manuscript’s rationale.

Methods

1, Consider making Table 2 supplementary information.

2, For the in vitro experiments where cytokines/chemokines such as TNF-α, IL-6, IL-12, and MCP-1 were assayed, please clarify if kits were used.

3, Specify how many in vitro experiments were performed and whether they were conducted in triplicate.

4, The classification of infection as Higher, Moderate, and Lower needs clarification. Please explain the method used and define each level.

Results and Discussion

5, Combine Figures 1 and 2 into a single figure, as Figure 1 by itself is not informative.

6, Adjust the order of Figure 2 to correctly present the classification of parasites according to LRV detection.

7, In Graph 3A, use different colors for each circle representing different parasites to show how each infects the hamsters.

8, Specify whether the L. (V.) guyanensis used was isolated from humans or insects.

9, Figures 3 through 7, which observe in vivo infection on different days, could be combined into one figure. Also, Figures 9 through 11 should be merged into one figure.

10, Figures 12 through 16 should be combined; also, clarify the letters used to symbolize statistical differences.

11, Address whether IHC was tested for granuloma regions, especially since granulomas were detected without the presence of parasites.

12, The Discussion section is overly exhaustive. Ensure that it is focused and directly addresses the main research question. The paragraphs are excessively long.

13, The aim presented in the Discussion seems to diverge from the rest of the manuscript. If the focus is on infection models, include more discussion on infection in THP-1 cells and consider the relevance of measured cytokines in these cells.

14, The Discussion redundantly revisits all results and includes unnecessary literature reviews. Streamline this section and focus directly on discussing the findings.

15, The conclusion primarily discusses L. (V.) lainsoni, which does not align with the stated major aim of the study. Please revise accordingly.

Minor comments throughout the manuscript:

Define ATL in the Introduction as well as in the abstract.

Line 99: There is no need to capitalize "leishmaniasis."

Lines 144-145: The mention of a wider project on the immunopathology of Leishmania parasites is unnecessary.

Line 179: Adjust to "Cells were exposed to L. (Viannia) species."

Line 181: Correct "Panoptic" usage.

Asterisks in Figures 4, 5, and 6 should be explained in the figure legends.

Line 316: The statement "During interaction with macrophages LPG and GIPLs had different levels of stimulation" is unclear.

Reviewer #2: (No Response)

Reviewer #3: The article is interesting, and the topic is quite relevant. However, important changes need to be made. The introduction does not make the objective of the article clear. However, both, the abstract and the author summary, suggest that the objective would be to establish an experimental model for Amazonian Leishmania (Viannia) species. It was unclear to me whether this was specifically the objective of the work, principally when we observe its title. I suggest that the authors rethink whether the article title could contain the possibility of establishing an experimental model for these species.

The introduction is disjointed and confusing. It needs to be reviewed. The objective of the work must be clear. The introduction needs to provide information so that the reader understands the importance of this objective.

Establishing an experimental model that allows us to unravel the mechanisms of the complex host-parasite relationship is essential. This knowledge can serve as a basis for, for example, developing new, more effective, and less toxic, chemotherapy drugs for leishmaniasis treatment. This could be one of the biases to be explored in the introduction, since the work points to immune mechanisms present in the parasite-host relationship, which could serve, in subsequent studies, as possible targets for immunomodulators to treat these parasitoses.

Required modifications

Introduction:

Line 94 - One sentence ends with the expression "depending on the region", and the subsequent sentence starts with the same expression. It is better to change the expression in the subsequent sentence. 

Lines 96-98 - Consider rewriting the sentence: “Recently, urbanization of ATL was reported in the Southeast Brazil, where L. (V.) braziliensis proven vectors Migonemyia migonei, Ny. intermedia and Ny. whitmani were detected in urban areas [9,10].”

Lines 101-102 - Consider rewriting the sentence: “Most LCL cases are in the north and northeast of Brazil, where L. (Viannia) parasites are endemic, the disease being a zoonosis and occupational hazard [12].”

Lines 107-108 - The sentence “A distinguished feature of this species is that it is often infected with an endosymbiont virus from the Totiviridae family named Leishmania RNA virus 1 (LRV1), known to increase disease severity.” is confusing. Consider exchanging for “The frequent infection of this species by Leishmania RNA virus 1 (LRV1), a Totiviridae virus known to increase disease severity, is a distinguished characteristic of L. guyanensis.”

Line 118 - Consider changing “At that time” to “Again”. 

Lines 123-126 - Consider rewriting the sentence: “A distinguished feature of L. (V.) lainsoni compared to the other L. (Viannia) species is that it possesses several biological, biochemical, and molecular characteristics that hinder its positioning within the same subgenus [25].”

PLOS authors have the option to publish the peer review history of their article (what does this mean?). If published, this will include your full peer review and any attached files.

Reviewer #1: No

Reviewer #2: No

Reviewer #3: No
---

## [Editor Report · Decision Letter 1]

28 Jun 2024

Dear  Dr. Rodrigo Soares,

We are pleased to inform you that your manuscript 'Unveiling the Enigmatic nature of six neglected Amazonian Leishmania (Viannia) species using the hamster model: Virulence, Histopathology and prospection of LRV1' has been provisionally accepted for publication in PLOS Neglected Tropical Diseases.

Best regards,

Ulisses Gazos Lopes

Academic Editor

Susan Madison-Antenucci

Section Editor

---

## [Editor Report · Acceptance letter]

4 Jul 2024

Dear Mr. Soares,

We are delighted to inform you that your manuscript, "Unveiling the Enigmatic nature of six neglected Amazonian Leishmania (Viannia) species using the hamster model: Virulence, Histopathology and prospection of LRV1," has been formally accepted for publication in PLOS Neglected Tropical Diseases.

Best regards,

Shaden Kamhawi

co-Editor-in-Chief

Paul Brindley

co-Editor-in-Chief
